# Icebergs, jigsaw puzzles and genealogy: Automated multi-generational iceberg tracking and lineage reconstruction.

Ben R. Evans[1], Alan R. Lowe[2], Anna Crawford[3], Andrew Fleming[1], J. Scott Hosking[1,2]

[1] British Antarctic Survey, Cambridge, CB3 0ET, UK
[2] Alan Turing Institute, London, NW1 2DB, UK
[3] University of Stirling, Stirling, FK9 4LA, UK

*Correspondence to*: Ben R. Evans (benevans@bas.ac.uk)

**Abstract.** Tabular icebergs calve from ice shelves and glaciers in Antarctica, Greenland, and northern Ellesmere Island. These 'ice islands', as they are referred to in the Arctic, drift, melt, and fragment, contributing freshwater and nutrients to the ocean, influencing circulation, carbon cycling and biodiversity in ways that remain poorly understood. Icebergs also pose risks to shipping, and maritime infrastructure. Improved understanding of iceberg drift and fragmentation will reduce uncertainties in

climate simulations and operational hazards. This study presents the first comprehensively validated, scalable multi-generational iceberg tracking approach and the first that is capable of reconstructing iceberg 'lineages' (here used to describe life histories including sources, where that source is a larger iceberg) through fragmentation events. This method enables a comprehensive reconstruction of iceberg paths from calving to their eventual disintegration, allowing for monitoring and source attribution across their life cycle.

We propose CryoTrack, an unsupervised approach based on iceberg geometry that is agnostic to data source or delineation method. The system requires only vector outlines. Initially, icebergs are linked across timesteps when their shapes remain similar, forming 'tracklets'. When significant shape changes occur, fragmented 'child' icebergs are linked to their 'parents' using a fuzzy geometric assembly method based on dynamic time warping, akin to assembling a jigsaw puzzle without image data. This approach reconstructs full iceberg lineages back to their calving origin. We evaluate system performance using

manually tracked iceberg outlines originating from Petermann Glacier and other northwest Greenland ice tongues. Standard tracking metrics and custom iceberg specific metrics assess its accuracy in scientific and operational contexts. Our approach achieves excellent tracking of icebergs with an overall tracking accuracy of 0.98 and 94% of iceberg area are correctly linked to source when icebergs are last observed.

This system, which focuses on the tracking of icebergs, but not the related and challenging problem of their detection,

contributes to the need for scalable iceberg monitoring. It enhances understanding of iceberg behaviours, impacts, and fragmentation, supporting process based and data driven predictive modelling for environmental and operational applications.

## 1 Introduction

Freshwater inputs to the oceans due to iceberg melting have the potential to influence ocean circulations, sea ice formation and nutrient and carbon cycles, with global environmental repercussions, yet iceberg dynamics and impacts are poorly represented

in numerical models due to a paucity of observations (Cenedese & Straneo, 2023). Iceberg flux represents roughly half of the total freshwater discharge from both the Antarctic and Greenland ice sheets (J. Bamber et al., 2012; J. L. Bamber et al., 2018; Coulon et al., 2024; Davison et al., 2020; Depoorter et al., 2013; Mottram et al., 2024). The locations of this freshwater input to the oceans can be far from the source location and substantially temporally delayed (Wagner et al., 2017), making this input difficult to quantify and model. For tabular Antarctic icebergs, 80% of ice loss has been shown to result from fragmentation into smaller icebergs, compared to 18% from basal melt (Tournadre et al., 2015). Being able to identify the source of large bergs and their fragments is therefore crucial to understanding the location and timing of most of the freshwater input to the oceans from icebergs. This capability would enable better parameterizations of freshwater distributions in ocean models (Huth et al., 2022; Marsh et al., 2015), improve their coupling to ice sheet models (Shiggins et al., 2023), aid evaluation of ecological impacts (Arrigo et al., 2002; K. L. Smith et al., 2013) and help mitigate hazards posed to humans, infrastructure and the environment (Fuglem & Jordaan, 2017; Hill, 2001; Mueller et al., 2013; Sackinger et al., 1985).

Icebergs are currently monitored by multiple national agencies for the provision of ice hazard information to marine stakeholders in the Arctic (e.g. Canadian Ice Service, International Ice Patrol), while the largest Antarctic icebergs (>18.5km in length) are tracked by the US National Ice Center. This tracking remains a largely manual endeavour. The requirement for substantial operator input limits current iceberg monitoring at both poles with restrictions to monitoring imposed based on geographical extents or iceberg size (e.g., Crawford et al., 2018a). Automated approaches to tracking will lead to more information being available to marine operators and will grow more extensive datasets for investigations into iceberg occurrence, drift and deterioration over time and space. As satellite technology improves, these automatically acquired datasets will also account for a greater proportion of the power law distribution that represents iceberg populations undergoing fragmentation (Crawford et al., 2018b; Enderlin et al., 2016; Tournadre et al., 2016). Such studies will furnish new insights to controls on motion (Crawford et al., 2016; Marson et al., 2018; Morison & Goldberg, 2012), fragmentation (A. J. Crawford et al., 2024; England et al., 2020; Huth, Adcroft, Sergienko, et al., 2022; Zeinali-Torbati et al., 2021)  and freshwater inputs (Crawford et al., 2018b; Huth et al., 2022; Stern et al., 2016). These advances will, in turn, support improved modelling of ice-shelf fracture and calving by enabling more comprehensive evaluation and validation. Improved representation of the processes and drivers of iceberg drift and deterioration will also further efforts to integrate process based and data driven models across the ice sheet-ocean interface, enhancing the fidelity of global climate models (Ackermann et al., 2024; R. S. Smith et al., 2021).

Advances have been made in automatic iceberg identification from satellite imagery in recent years (Barbat et al., 2019; Moyer et al., 2019; Shiggins et al., 2023), though most approaches are not yet sufficiently scalable to support operational monitoring (Evans et al., 2023) and developments in this field are ongoing. While iceberg detection is a necessary step, our work focuses specifically on the downstream task of tracking icebergs once they have been detected in a time series of satellite images. Previously Barbat et al. (2021) developed an automated approach for tracking icebergs present in satellite scenes of the Weddell Sea. That approach relied on Jaccard similarity between shape descriptors, principally a vector of radial distances from centroid

to perimeter. They used the tracked icebergs to infer drift and melt rates but did not attempt to link across fragmentation events.

Indeed, they observed that their tracker's principal failure mode was when fragmentation or large melt events occurred, although they did not offer a comprehensive evaluation of the tracker's characteristics. Koo et al. (2023) used similar shape descriptors to track icebergs detected by their algorithm but did not present a substantial evaluation. Earlier attempts at tracking have also been made (e.g., Silva & Bigg, 2005) but no studies have yet tried to reconstruct lineages starting from an iceberg's source location and spanning fragmentation events. The majority of smaller (yet often still tabular) icebergs are calved from

larger icebergs rather than directly from ice shelves (Tournadre et al., 2016). Understanding the sources and fates of these fragments of larger icebergs is therefore a critical aspect of understanding freshwater fluxes and distributions. This study addresses some of the challenges to better understanding the impacts of icebergs on the global system by presenting the first comprehensively evaluated, automatable and scalable iceberg tracking methodology of which we are aware, and also the first iceberg tracking schema capable of maintaining lineage associations between icebergs across fragmentation events.


Tracking of icebergs sits within the broad domain of Multiple Object Tracking (MOT) problems. Most MOT methods are based on tracking unchanging objects in sequences of natural images and transformer architectures have recently been widely employed to produce state-of-the-art (SOTA) trackers (e.g., Chu et al., n.d.; Meinhardt et al., n.d.; Sun et al., 2020). The iceberg tracking problem is, perhaps, most similar to the problem of tracking cells in live cell microscopy data since both contexts

must be able to handle division of objects (fragmentation for icebergs / mitosis in the context of cells), as well as movement, changes in shape and other attributes, and disappearance (melt / apoptosis). Cell tracking is a well developed field (Ulman et al., 2017) with transformer based architectures also recently achieving SOTA performance. Gallusser & Weigert (2025) recently proposed the first transformer tracking approach that is capable of handling division events. Nevertheless, and irrespective of architecture, we are not aware of any tracking approaches explicitly designed to be capable of handling division

into more than two child objects, which is necessary for tracing the lineage of large tabular icebergs that may experience large fragmentation events that produce many child icebergs.

The iceberg tracking problem is further differentiated from other tracking challenges by the geospatial context, topological constraints, and complex environmental fields (wind, currents, sea ice concentration and drift etc.) that dictate iceberg

movement. Additionally, the objects to be tracked vary dramatically in size. The surface area of tabular icebergs tracked in the the Canadian Ice Island Detection, Drift and Deterioration (CI2D3) Database, upon which we base this study, vary by 5 orders of magnitude (Crawford et al., 2018a). Their highly variable observed mobility, coupled with a sparse and irregular sampling frequency (relative to laboratory or video based sequence acquisitions available in microbiological studies) further exacerbates the tracking challenge for icebergs since they can move by hundreds of kilometres between observations to be well outside

their previous footprint. There is also a pervasive missing data problem that arises from satellite acquisition schedules and meteorological conditions when constructing image sequences. Most MOT and cell tracking methods proposed to date are also supervised in nature and therefore require extensive datasets of manually labelled pairs of image and segmentation mask to

learn object associations. While the CI2D3 Database (Crawford et al., 2018a) that we use to develop our presented approach contains numerous segmentations, the underlying image data are not available to the authors for the purposes of this study,

and we are not aware of any suitable annotated datasets upon which to train a supervised method. The approach proposed here is therefore fully unsupervised, which offers advantages for transferability across geographical contexts and data modalities. We employ tools and evaluation metrics developed for live cell tracking contexts but introduce a novel geometric assembly process along with evaluation metrics tailored to the expected downstream applications.

## 2 Data

We use the CI2D3 Database to develop and evaluate our proposed method. While other iceberg databases exist (e.g. Brigham-Young University / National Ice Center ((Budge & Long, 2018)), the CI2D3 Database is, to our knowledge, unique in containing comprehensive lineage information for icebergs down to, at times, 0.1km² in areal extent. The CI2D3 Database contains more than 25000 polygons, manually delineated from a combination RADARSAT-1 and -2 and Envisat imagery selected with a target revisit period of two weeks, representing large, tabular icebergs ("ice islands") that originated from

calving events at the Petermann Glacier, northern Greenland in 2008, 2010, 2011 and 2012, along with calving events from other floating ice tongues in that region (Crawford et al., 2018a). Lineage associations were manually ascribed by the expert annotator, taking into account proximity, shape and appearance including surface patterns and textures. While manual determination of lineages implies a degree of uncertainty, it represents the most reliable method available. Nevertheless, the reference dataset's limitations will affect the tracking results. For example, we have observed that at least one iceberg with

near identical geometry and close proximity that we believe to be the same iceberg, but which lacks a track linking the observations in the CI2D3 dataset. Such artefacts of the manual annotation process are believed to be rare but have the potential to affect the performance metrics for our automated tracking approach.

## 3 Methods

We adopt a tracking-by-detection approach to the problem, as is common across many MOT domains (Gallusser & Weigert,

2025). Within this framework, objects are initially segmented in a detection step before being tracked in a secondary step. In the case of manual delineations, as conducted for the generation of the CI2D3 Database, detections are in polygon (vector) format denoting the perimeter of the icebergs. Automated iceberg detection approaches vary but tend to produce segmentation masks representing presence or absence of iceberg on a per pixel basis. These can easily be converted to polygons. Some object detection methods may return properties of the identified regions (icebergs) such as texture or intensity, while others may

return deep feature embeddings. However, these additional properties are not always available and would not be consistent across source data modalities. The tracker we propose here is therefore designed to operate on the lowest common denominator information supplied by all detection workflows, namely the geometry of each detection. This means it is highly generalizable

and agnostic to the process that generates the iceberg segmentation. The tracking process consists of five stages: data preparation, tracklet construction, generational linking, lineage reconstruction and evaluation.

## 3.1 Data preparation

The contents of the CI2D3 Database are shown in Figure 1. We selected a spatial subset for development and evaluation that contains the calving tongue of the Petermann Glacier, the source of most of the icebergs in the dataset. The subset (delineated red in Figure 1) encompasses any icebergs from Petermann Glacier and those drifting from more northerly glaciers as they follow the prevalent drift pattern to the south through the Nares Strait. As such, the spatial extent of our subset encompasses the source of most icebergs and the densest field of observations in the dataset and should present the most challenging environment in which to track icebergs because it contains the largest numbers of spatially close and contemporaneously observed icebergs, as well as the largest numbers of the smallest icebergs.

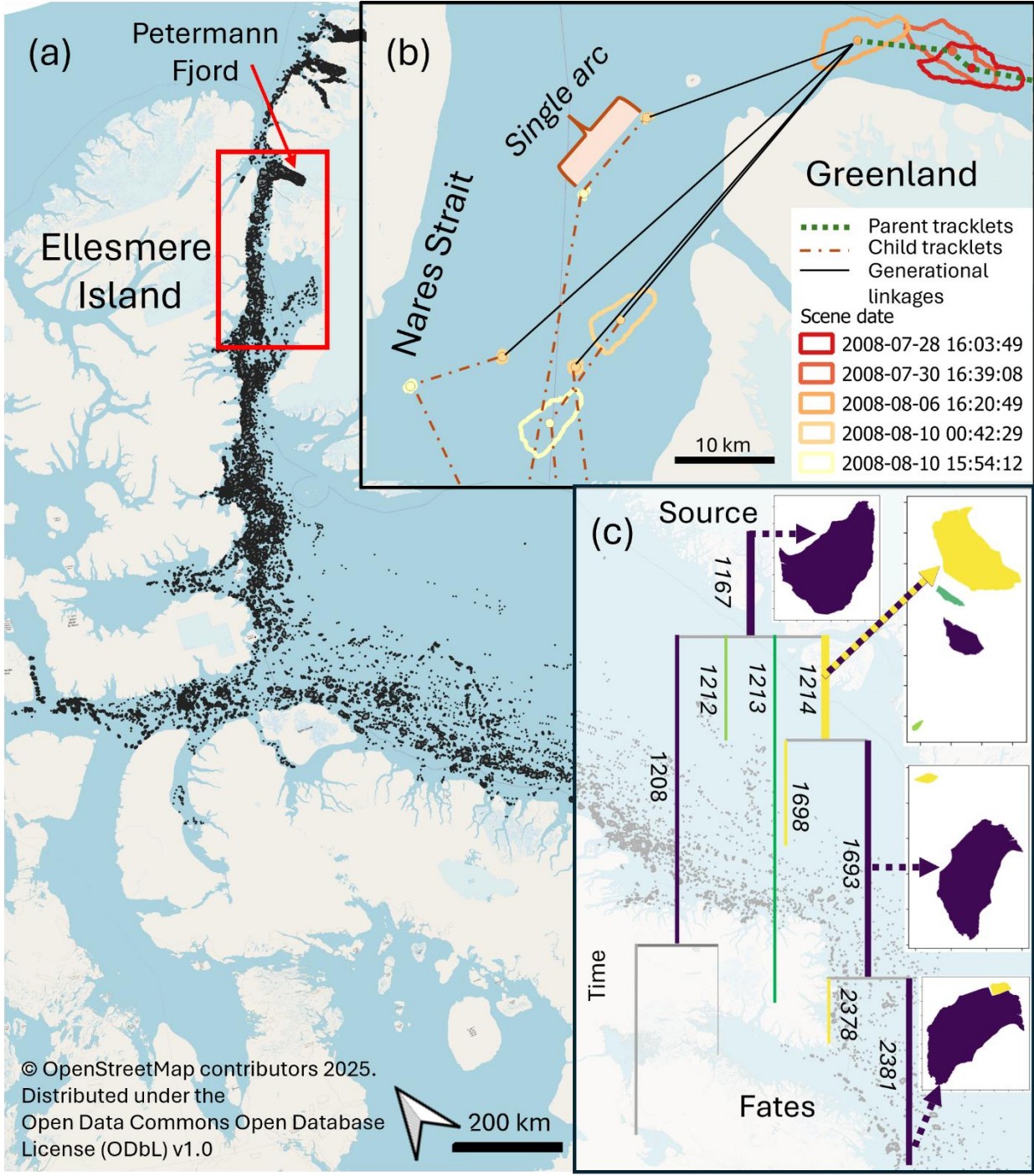

**Figure 1:** **(a) Detections (black) in the CI2D3 Database spanning 2008-2013 with the spatial subset used here defined by the red box. (b) Example of tracklets and generational linkages for part of an iceberg lineage. (c) Schematic of partial lineage tree representing the fragmentation of an iceberg (ID 1167) within the CI2D3 Database, following the branches containing the largest fragment at each division. Colours of branches correspond to the iceberg outlines on the right, numbers denote iceberg ID. Map data: https://www.openstreetmap.org/copyright.**


Within the CI2D3 Database, each iceberg observation has a unique identifier, with lineage information contained in a field denoting, in the case of drift, the identifier associated with the previous observation of that iceberg or, in the case of fragmentation, the identifier will be that of the parent iceberg prior to fracture. This representation was initially converted for this study such that an iceberg retains the same unique (integer) identity across time points unless it divides into two or more fragments, at which point each child iceberg is assigned a new unique identifier and a 'parent' attribute denoting the ID of the

iceberg that fragmented to form it ((c), Figure 1).

The domain contains multiple satellite scene footprints. Each observation timepoint, therefore, does not provide full coverage of the entire domain (even for the subset used in this study) and the remainder is effectively missing data. As such, the absence of icebergs in the missing data region does not imply an absence of icebergs at that point in time, merely an absence of

observations. For any given location, therefore, the observations are temporally sparse relative to the overall sequence of all observation timepoints that comprise the whole domain. The target observation interval for any given point in the CI2D3 Database was two weeks. For the purposes of demonstrating the proposed method, the dates at which any observation was contained in the database were stacked and a uniformly incrementing timestep assigned to that date, implying that the physical time interval between successive timesteps is non-uniform. For the test subset area this resulted in 706 observation timepoints

between 2008 and 2013. We recognize that this simplistic treatment of the time domain presents issues but the development of a more general schema for simultaneously handling spatial and temporal sparsity within tracking problems is beyond the scope of the current work. Each polygon in the CI2D3 Database is represented by its geometry, which we resampled to a uniform 256 vertices equally spaced around the perimeter (see codebase for implementation), and has attributes of its own identity ('ID'), its parent's identity ('parent') and the timestep ('t') in which it was observed. In addition to these, the original

iceberg to which each can track its lineage through its parents is denoted by a 'root' attribute. The 256 vertex resampling ensures that, even for very large icebergs, the outline alignment stage (3.3.1 Outline alignment:) remains computationally tractable, which would not be guaranteed if using a uniform-distance resampling or without resampling at all. Furthermore, resampling to a uniform number of vertices helps to propagate some scale awareness to the amplitude component of the 1-d distance vectors (Figure 2b) upon which iceberg associations are based, helping to exploit information on the relative sizes of

the iceberg when proposing matches.

## 3.2 Tracklet construction

The tracklet construction stage is analogous to the tracking approaches described in previous studies (Barbat et al., 2021; Koo et al., 2023; Silva & Bigg, 2005). In this stage, icebergs that do not change shape substantially between observations are linked,

as illustrated by the dashed lines in panel B of Figure 1, where a tracklet refers to the path of a single iceberg, potentially across multiple consecutive observations. A path covering a single time step within a tracklet or generational linkage is referred to as

an arc. The method must be able to associate icebergs that change slightly through time as they melt and small parts (below the detection limit) calve. We take a conceptually similar approach to that proposed by Barbat et al. (2021) in that we build associations between icebergs based on their size and shape. We derive five features to describe each shape. We use three simple features, namely area, length and perimeter. We use an additional two features to describe the complex geometry of the icebergs (UMAP-1 and UMAP-2). To compute these, we fit $10^{th}$ order elliptical fourier descriptors (EFDs, Kuhl & Giardina, 1982) to the perimeter shape, implemented using the pyefd python package (https://github.com/hbldh/pyefd, 2024). This results in 40 coefficients that are normalized to be rotation and translation invariant, but not size invariant. We then use a UMAP dimensionality reduction (McInnes et al., 2018) to reduce this to the two additional features. All five features are rescaled 0-1. We then use Bayesian Tracker (Btrack, Ulicna et al. (2021)), a python package developed for live cell tracking, to establish tracklets for which geometric characteristics do not change dramatically (i.e. they are similar enough that Btrack can recognise them as the same iceberg across successive observations). We use the 'visual features' linking but disable the motion model that places spatial priors on future iceberg locations since it is poorly suited to predicting the highly variable movement of icebergs and the non-uniform time spacing of observations. We also do not conduct global optimization, the step in which Btrack attempts to construct links between tracklets and establish parent-child relations since the heuristics are not appropriate for the iceberg context (see introduction). In the process of tracklet generation, Btrack constructs a Bayesian belief matrix for each timestep with uniform prior and dimensions N x (M+1), where N is the number of existing tracks and M is the number of objects detected in the current field of view. Bayesian updates are then performed based on cosine distances between the feature vectors for all pairs of icebergs within a given search radius of each other to calculate the probability of a link being established or the iceberg being considered lost (by reference to a tuneable parameter, see config file). Finally, iceberg associations are chosen, given the belief matrix, based on the maximum posterior probability of either an association or loss of the tracklet. Icebergs in the current frame that have not been associated with an existing tracklet generate a new tracklet while lost tracklets persist as dummies for a prescribed number of timesteps (see below). Using the five visual features, the median cosine distance between icebergs and other temporal instances of the same identity was $3.2E^{-9}$, whereas the median distance to the icebergs with a different identity was seven orders of magnitude larger at 0.05. This indicates effective separation of geometries in this 5-dimensional feature space. To handle the temporal data sparsity problem arising from the large domain and intermittent satellite coverage of any one location within it, Btrack is able to insert dummy instances for a prescribed number of timesteps between linked observations. If an iceberg is not observed again within the given time buffer the tracklet is terminated. The search radius and time buffer are tunable parameters that were set, through experimentation, at 100km and 6 timesteps respectively. Optimal values of these will be a function of the domain extent, data frequency and environmental factors controlling iceberg motion. Increasing them will tend to increase the false positive linkage rate while decreasing them will tend to increase the false negative rate. Ulicna et al. (2021) provide a detailed explanation of how Btrack constructs tracklets, and the reader is referred there for further detail. The configuration file for the Btrack step is available alongside the codebase (see code availability).

## 3.3 Generational linking

Generational linking matches 'child' fragments to their 'parent', which is a larger iceberg, as shown as solid black lines in (b) of Figure 1. This is achieved through a process of tessellating child fragments within the outline of the parent iceberg in a manner similar to assembling a jigsaw puzzle (Zhang et al., 2017) but without any image information to assist and in the presence of the potential for substantial portions of the parent to have been lost entirely from the detections due to melt and small scale fracture. We use this process to assess which shapes share similar parts of their geometries and between which it is possible to make legitimate parent-child linkages. The challenge is to match the high frequency components of the perimeter shape while ignoring the global invariances of translation and rotation that arise from iceberg drift between observations. Furthermore, due to melt and small scale calving (below the detection limit) modifying the edges of icebergs, imperfect segmentation recall, and sub-pixel uncertainties in edge position, it is unlikely that there will ever be perfect correspondence between any parts of the perimeter shapes of parent and child icebergs. Similarly, it is unlikely that the total area of children emanating from one parent will exactly match the original area of that parent.

### 3.3.1 Outline alignment:

The core of the process is an outline alignment step, whereby sub-sections of shape perimeters that are similar between icebergs are used to align potential children to potential parents (Figure 2).

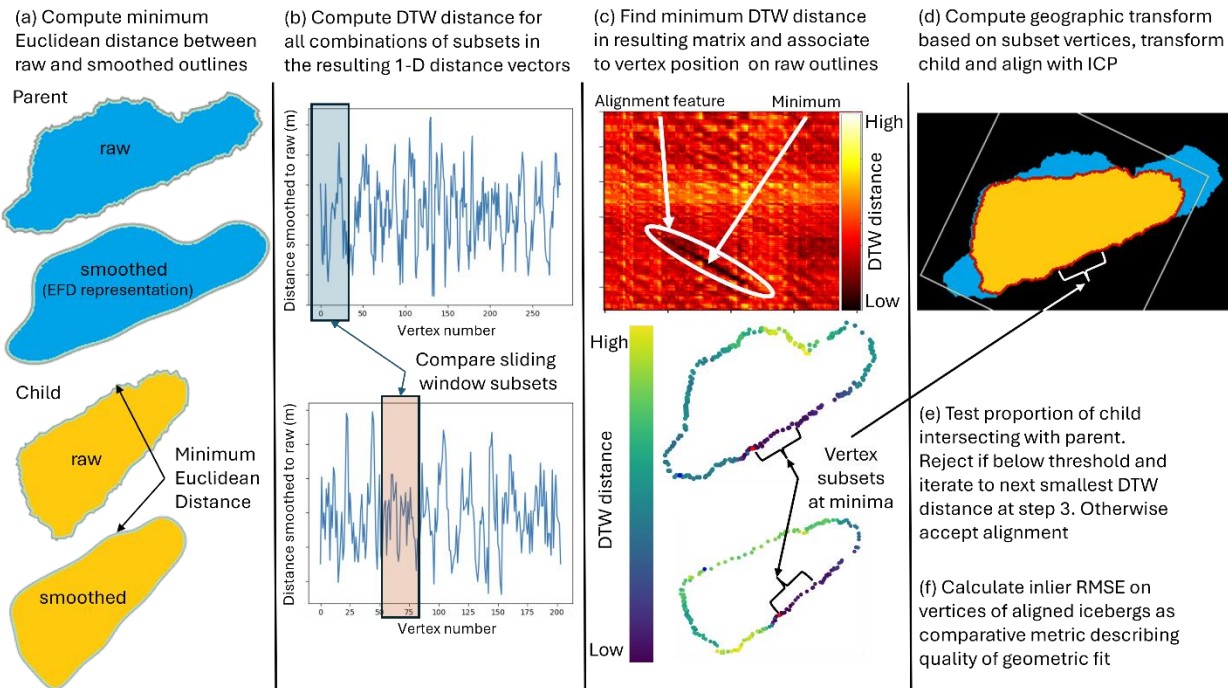

**(a) Compute minimum Euclidean distance between raw and smoothed outlines**

**(b) Compute DTW distance for all combinations of subsets in the resulting 1-D distance vectors**

**(c) Find minimum DTW distance in resulting matrix and associate to vertex position on raw outlines**

**(d) Compute geographic transform based on subset vertices, transform child and align with ICP**

**(e) Test proportion of child intersecting with parent. Reject if below threshold and iterate to next smallest DTW distance at step 3. Otherwise accept alignment**

**(f) Calculate inlier RMSE on vertices of aligned icebergs as comparative metric describing quality of geometric fit**

**Figure 2: Process for outline alignment of child with parent based on finding minimal dynamic time warping (DTW) distances between high frequency components of perimeter shapes – Represented as Process 4 in Figure 3.**

To isolate the high frequency shape components and remove translation and rotation we first smooth the raw perimeter of each

shape using a $5^{th}$ order EFD and reconstruct the shape from the coefficients and centroid. We then take the Euclidean distance from each vertex in the raw perimeter to the nearest point on the smoothed outline. Distances are negative where the raw outline is further from the centroid than the smoothed outline ((a), Figure 2). This produces a 1-dimensional (1-D) vector of deviations between raw and smoothed outlines. We then use dynamic time warping (DTW) to estimate similarity between subset regions of these 1-D vectors using a sliding window approach ((b), Figure 2). DTW is a curve matching algorithm that

estimates dissimilarity between sequences as a warping distance, which is low when sequences align well and high when they align poorly. It is widely used in audio, speech and text recognition (Müller, 2007; Myers & Rabiner, 1981) and does not assume correspondences between the vertices of the two sequences. For each pair of sub-sections (in our case each 10 vertices long), we compute a DTW distance using the *dtaidistance* Python package (Meert et al., 2020), producing a matrix of DTW distances, in which areas where the shape perimeters align well are observable as minima ((c), Figure 2). We take the sliding

window subsets corresponding to the lowest DTW distance found and use the geographic coordinates of the vertices to compute a least-squares transformation matrix between them. We apply this transformation to the child iceberg to translate and rotate it, thereby superimposing it on the parent iceberg ((d), Figure 2). We then perform an iterative closest points alignment on all vertices of the parent and aligned child to reconcile any small positional errors. These largely arise from angular errors in the transformation estimation. We impose an experimentally determined heuristic constraint that the alignment must result in more

than 96% of the area of the child being within its intersection with the parent ((e), Figure 2). This constrains children to fall largely within the parent geometry. If this constraint is breached we discard the alignment and iterate to the sub-sequences with the second smallest DTW distance, repeating the transformation and overlap checks. This process is repeated for DTW distances below the median of the matrix until a satisfactory alignment is found. If no alignment is found the child is not linked to the parent. Having accepted an alignment we compute the inlier RMSE (root mean squared error) of the vertex coordinates

to represent how good the geometric fit between the outlines is and upon which to compare competing possible alignments ((f), Figure 2).

### 3.3.2 Tessellation

At any given time step there may be multiple potential parents and children. The alignment process described above for a

single parent-child linkage is deployed within an iterative workflow in such circumstances in order to tessellate multiple children within parents (Figure 3).

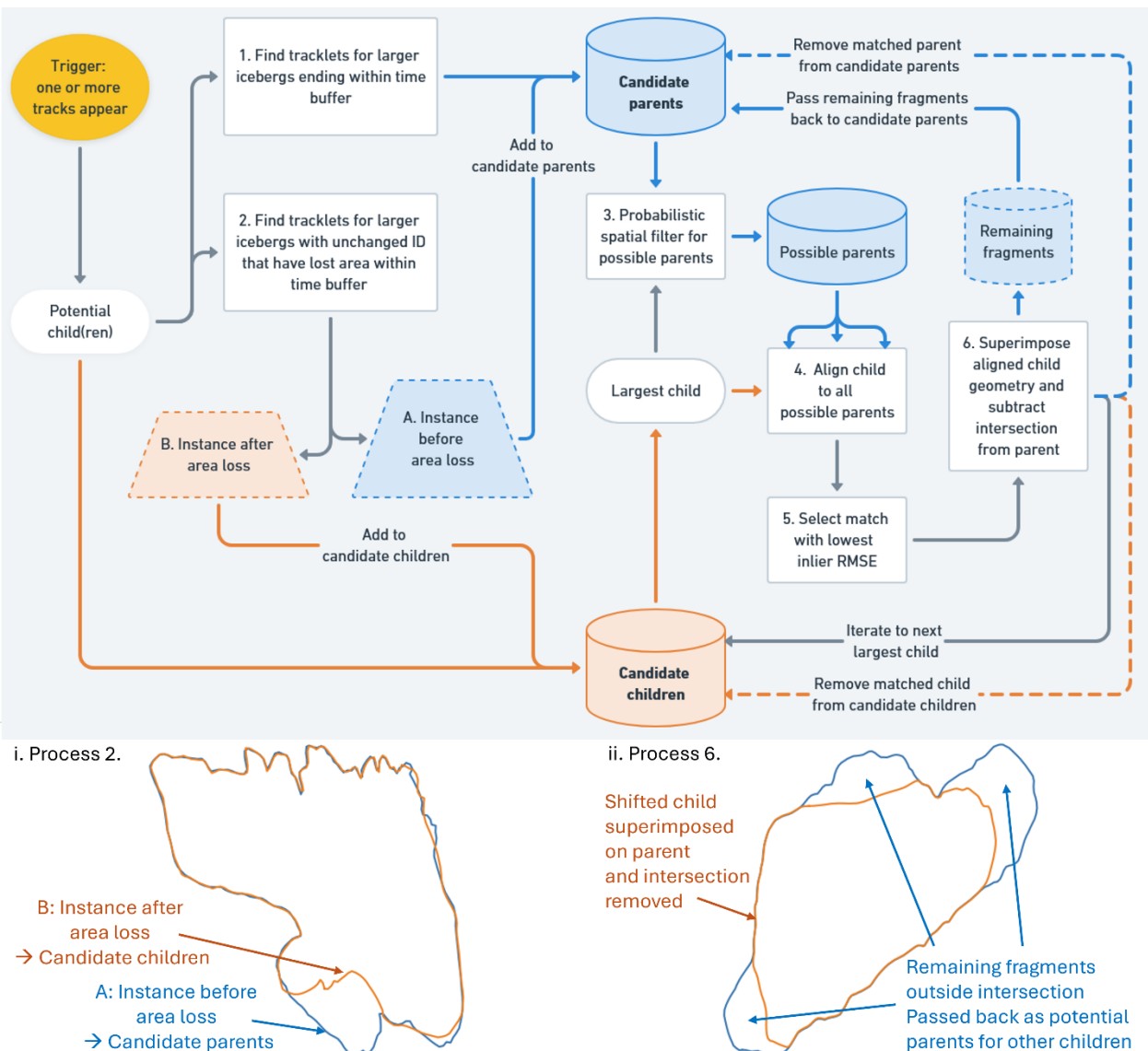

**Figure 3: Tessellation workflow for associating multiple parent and child icebergs. Blue denotes parents, orange denotes children. Processes are shown in rectangular boxes. Insets i. and ii. illustrate Processes 2 and 6 respectively. Process 4 corresponds to the outline alignment represented in Figure 2.**

The workflow is triggered when a previously unseen iceberg appears (i.e. a new tracklet is initiated). In such situations we require an explanation for the appearance of an iceberg that we have not previously observed and calving from a larger iceberg is the most probable explanation (Barbat et al., 2021), particularly when far from glacier or ice shelf calving fronts (we discuss the limitations of this assumption further below). The potential source could either be an iceberg that has disappeared (a tracklet that has ended, see process 1, Figure 3) or an iceberg that continues to be observed but has lost sufficient area to account for the newly observed iceberg (process 2, Figure 3). In the latter case, the most recent previous observations of such

icebergs are treated as a candidate parents (A, blue, in inset i. of Figure 3) while the corresponding iceberg at the same time point as the track appearance trigger becomes an additional candidate child (B, orange, in inset i. of Figure 3) such that its intersection is removed from the parent A following Process 6 (Figure 3) before testing the newly appeared iceberg for fit to any fragments that remain.

We thus end up with a list of candidate parents and a list of candidate children. Starting with the largest candidate child, we identify possible parents within the preceding time range using a probabilistic spatial filter (process 3, Figure 3) based upon vector fields interpolated from the tracklet data (see Appendix A: Probabilistic spatial filter for constraining possible parent icebergs:). In contrast to the fixed search radius of 100km used for tracklet construction, this allows us to inform where we look for matches based upon the tracklet observations and the time interval between observations, helping to constrain the locations of proposed generational linkages to be consistent with the observed motion of icebergs between fragmentation events. We perform alignment (Figure 2 and process 4, Figure 3) against all possible larger parents. We take the alignment with the lowest inlier RMSE following iterative closest point (ICP) registration as being the most likely parent-child relationship (process 5, Figure 3). We then remove the intersection of the aligned child's geometry (orange in inset ii. to Figure 3) from that of the parent (blue in inset ii. to Figure 3). This leaves parts of the parent unaccounted for, from which other children could be derived (inset ii. Figure 3). These remaining parts are added back to the candidate parents list while the aligned child is removed from the candidate children list and its parentage recorded and the parent is removed from the candidate parents list, having been accounted for. This process is repeated until all candidate children have been assigned to a parent or there are no more valid alignments found. Candidate children for which no alignment to a parent is found initiate a new lineage.

We thus have five tuneable parameters within the generational linking stage. Sigma is the standard deviation of the gaussian used to model spatial source probabilities while probability threshold determines the probability above which the location of a candidate parent is accepted for consideration in generational linking (see Appendix A). Proportional overlap is the proportion of the child's area that must fall within the parent for the match to be deemed valid, time buffer is the maximum number of timesteps over which matches are considered and sub-section lengths is the number of vertices used to calculate DTW distances.

### 3.3.3 Lineage reconstruction

The tessellation procedure is conducted across the dataset, iterating by time step. We then enforce rules about how lineages are represented. Icebergs maintain a single identity for as long as no fragmentation event is detected. A fragmentation event is defined as when two or more icebergs share a parent. Consequently, an iceberg may change shape and size substantially while maintaining an identity if no others can be aligned to the parts it loses. Conversely, it may remain substantially the same shape,

but if a small fragment calves and is associated with it (as in Figure 3, panel i), both that fragment and the largely unchanged iceberg will be assigned new identities (thereby initiating new tracklets) and their parent attribute set to the initial identity. Parents may be linked to many children but a child may only be linked to one parent. We are thus able to reconstruct the lineage trees of icebergs (Panel C, Figure 1) in an automated fashion for the first time. We discuss the limitations and uncertainties that arise, along with further work required to improve the performance of this step below.

### 3.3.4 Evaluation


To our knowledge, object movement and lineage tracking have not been previously explored in a geospatial context, nor in cases where track branching may result in more than two children, as seen in cell tracking. Consequently, there are no established performance metrics for our context. However, we adapt metrics from the cell tracking domain to assess CryoTrack's performance. We used the *traccuracy* python package (github.com/live-image-tracking-tools/traccuracy), with a
custom data loader to handle geospatial vector formats, to evaluate our outputs against the manually ascribed lineages encoded in the CI2D3 dataset (GT).

We report three transferrable metrics derived from the Cell Tracking Challenge (Ulman et al., 2017): Tracking Accuracy (TRA), Target Effectiveness (TE) and Track Purity (TP). TRA describes how well all objects (icebergs) are both identified
and tracked (although in this case there is no detection step). TE describes the proportion of each reference track for which the longest reconstructed track overlaps, averaged over all reference tracks. TP is the inverse of TE, being the proportion of each reconstructed track for which the longest reference track overlaps, averaged over all reconstructed tracks. All three vary in the range 0-1, with 1 being perfect reconstruction of the tracking graph. The reader is referred to Matula et al. (2015) for further detail. In addition to these, we introduce new evaluation metrics tailored to scientific and operational applications of iceberg
tracking.

Scientific applications focus on identifying iceberg origins, reconstructing drift trajectories, determining fragmentation timing, and quantifying area loss rates over long timescales, potentially spanning years or decades. Performance in this context depends on whether an iceberg can be correctly linked back to its original source, regardless of where or when it is observed. We define
Root Precision (RP) as the proportion of icebergs correctly attributed to their source at their last observed position. Root Area Precision (RAP) extends this by weighting RP according to iceberg area, emphasizing the accuracy of total ice mass attribution. Operational applications focus on hazard avoidance (Smith et al., 2025), where the priority would be accurately tracking icebergs over shorter timescales (days or weeks) to infer recent trajectories and predict future locations over relatively short timescales. To assess performance in this context, we evaluate how well predicted tracks match ground truth tracks over
different time intervals. We report precision (true positives divided by all positives), recall (true positives divided by the sum of true positives and false negatives), and F1-score (harmonic mean of precision and recall) for different lead times, illustrating the reliability of tracks, and therefore trajectories, over those intervals.

Generational (parent-child) linkages between icebergs are assessed based on their agreement with the ground truth dataset. Since icebergs can divide into more than two fragments, these relationships are evaluated independently rather than requiring a strict two-child split, as in cell tracking literature. Generational linkages in the predicted set may also be represented by tracklets in a continuous track in the GT set, and vice versa. Such linkages themselves are also treated as true positives since they link the correct two objects, although they do imply either commission or omission of another generational linkage at the same stage. Division Precision (DP), Division Recall (DR), and Division F1-score (DF-1) measure the accuracy of these generational linkages.

We anticipated tracking to be most challenging in the congested areas close to the calving front of Petermann Galcier. This is particularly true because the dataset currently does not allow for the glacier to be represented as a potential source of newly observed icebergs. To investigate the effect of near-glacier confusion we also evaluated performance for a subset that excludes the fjord (see Appendix B: Exclusion of fjord for method).

## 4 Results

### 4.1 Iceberg lineages:

We tested a variety of combinations of parameters for the generational linking stage, observing the expected trade-offs between precision and recall as we varied the effective search radius defined by the sigma and probability threshold parameters (larger search domain increases precision and decreases recall, and *vice versa*). Allowing lower proportions of overlap when matching shapes leads to less well constrained matches, reducing precision while meaning that the shapes of remaining fragments for tessellation of smaller icebergs are less robust, decreasing recall. Lengthening or shortening the time buffer tends to decrease precision but is a function of the temporal sparsity of observations in the domain so is informed by the dataset structure. Lengthening the sub-section length for the DTW distance matrix comparison adds computational complexity and reduces performance for the smaller icebergs with fewer perimeter vertices while shortening it reduces the information available for DTW calculation too much. We did not observe any extreme, abrupt or unexpected sensitivity to any of the configurable parameters during our tests. The final configuration for which we report performance used the following parameters: sigma = 5000 m; probability threshold = 0.05; proportional overlap = 0.96 (corresponding to the 96% threshold described above); time buffer = 6 timesteps; sub-section length = 10 vertices. Figure 4 shows examples of lineages reconstructed using our method are shown for two timepoints (t=341 in main panel and t=370 in inset) to illustrate correct tracks and various possible failure modes. Five points of interest (A-E) are marked. Point A shows a fragmentation event that is identified by white circles on both panels to aid orientation. This event produced two FP generational linkages that also imply two FN arcs. B marks a single FN arc in an otherwise long and correct track for a small iceberg. C marks the fragmentation shown in more detail in Figure

365    5c(iii) where three children are correctly matched and one missed. D marks a correctly tracked fragmentation into two children and E shows successive failures (both FP and FN) in the track of a very small iceberg.

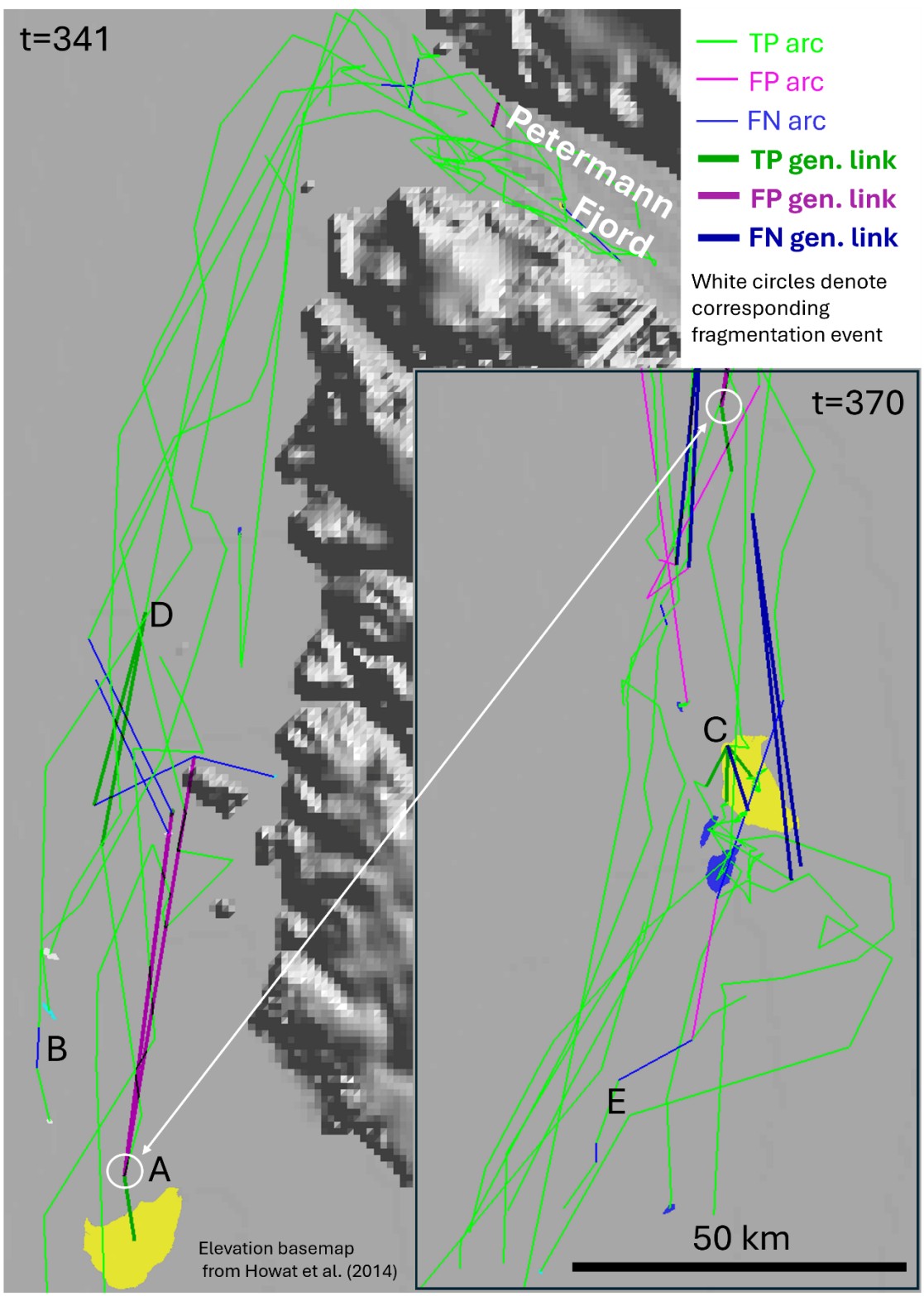

t=341

t=370

Petermann Fjord

TP arc
FP arc
FN arc
**TP gen. link**
**FP gen. link**
**FN gen. link**

White circles denote corresponding fragmentation event

D

B

A

C

E

Elevation basemap from Howat et al. (2014)

50 km

**Figure 4: Example tracks reconstructed from the CI2D3 dataset prior to two timepoints (t=341 in main panel, t=370 for inset). True positive (TP), false positive (FP) and false negative (FN) arcs and generational links are shown. Point A (white circle) denotes a fragmentation resulting in one TP generational link and two FP generational links. This event is identified in both pan els for orientation purposes. B denotes a FN link in an otherwise long correct track, C corresponds to the fragmentation shown in Figure 5c(iii), D marks a correctly tracked fragmentation, E denotes a very small, poorly tracked iceberg with both FP an FN arcs Greenland elevation data from GIMP-DEM 90** (Howat et al., 2014).

## 4.2 Performance:

Performance, as evaluated against the metrics described in Sect. 3.3.4, is reported in Table 1. The tracker exhibits strong performance overall, with tracks closely reflecting the manually annotated ones with high overall accuracy and long periods of perfect track overlap, particularly between fragmentation events. Fragmentation is captured less well but demonstrates good performance given its novelty and presents clear avenues for future improvement.

**Table 1 - Tracker performance**

| Metric | Full study domain | Domain excluding fjord |
|---|---|---|
| Tracking Accuracy (TRA) | 0.98 | 0.99 |
| Target Effectiveness (TE) | 0.72 | 0.83 |
| Track Purity (TP) | 0.87 | 0.88 |
| Root Precision (RP) | 0.51 | 0.61 |
| Root Area Precision (RAP) | 0.94 | 0.96 |
| Division precision (DP) | 0.38 | 0.70 |
| Division Recall (DR) | 0.35 | 0.34 |
| Division F1-score (DF-1) | 0.37 | 0.46 |

The discrepancy between RP and RAP arises from the size distribution of icebergs within the dataset and differential tracking performance for different sized icebergs. The relationship between RP and the size of the tracked iceberg is illustrated in Figure 5a (blue bars), where the grey histogram illustrates the frequency of icebergs within each size class. Icebergs are grouped by order of magnitude of surface area, an approach that reflects the size categories proposed by Wesche & Dierking (2015). RP is high for the larger size classes, decreasing as iceberg surface area declines.

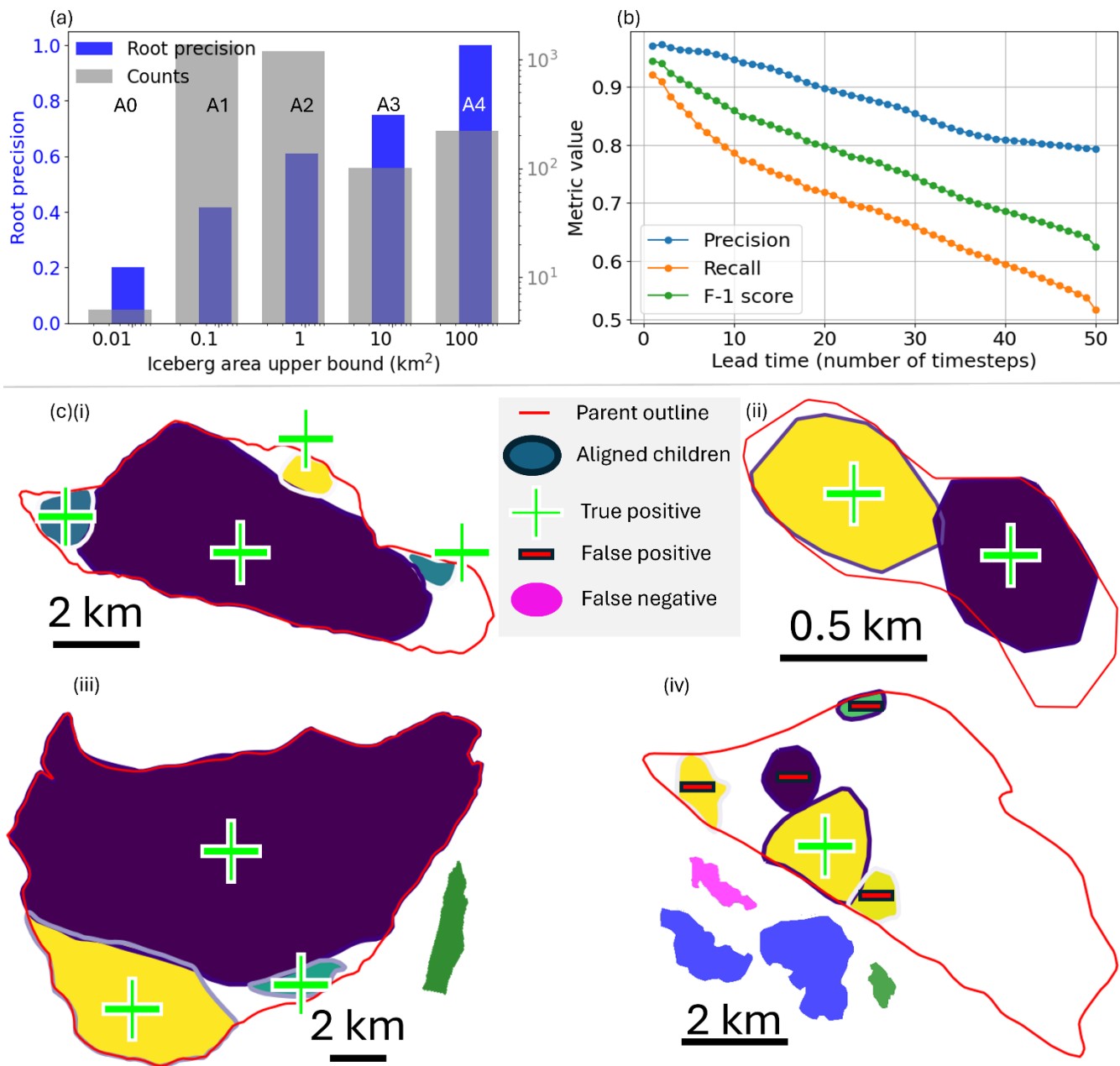

**Figure 5: (a) Root precision (blue) by size class of iceberg with size classes A0-A4 following** Wesche & Dierking (2015)**, histogram of iceberg observations (grey). (b) Performance in maintaining iceberg identity over 50 non-uniform time intervals. (c) examples of automated tessellations (arbitrary scale and colours with contrasting outlines to illustrate where fitted shapes overlap. True positive (green plus) and false positive (red minus) associations are indicated. Shapes without outlines that fall outside the red parent outlines for (iii) and (iv) are false negative associations.**

Performance, as it relates to navigational uses, was strong, with precision, recall and F1-score for maintaining correct iceberg identities across all single observational time intervals of 0.97, 0.90 and 0.93 respectively. Performance remains strong, with
F1-score exceeding 0.75, out to lead times of 30 time steps as shown in Figure 5b.

Performance in establishing generational linkages is weaker than for other aspects. Our geometric assembly method achieved a Division Precision of 0.39, Division Recall of 0.35 and Division F1-score of 0.37. Overall performance improved when the fjord area was excluded, this being driven by a substantial increase in DP to 0.70, bringing DF-1 to 0.46 despite a small drop
in DR (0.34). Examples of tessellations enabling reconstruction of complex many-to-one generational associations are shown in Figure 5c illustrating various success and failure modes. Panels i and ii show wholly correct tessellations for medium and small parent icebergs respectively. Panel iii shows a large parent iceberg with three correct child linkages but one false negative association (shown outside parent outline) that was not made. Panel iv shows a largely incorrect set of linkages with only one child correctly attributed and four false positives and four false negatives.

**5 Discussion**

Our proposed method exhibits good performance when evaluated using metrics derived from the Cell Tracking Challenge (CTC) (Ulman et al., 2017). The TRA performance of 0.98 is artificially elevated since the metric includes a component of detection performance. We use the same detections for tracking and evaluation, which implies perfect detection. Nevertheless,
this metric may serve as a useful benchmark for future studies applying similar methodologies to tracking objects in machine learning derived segmentations for which independent reference data are available. The values of TE and TP (0.72 and 0.87 respectively) imply that we typically achieve overlap between reconstructed and reference tracks for substantial portions of their lengths.

Our custom metrics derived to support expected scientific downstream applications (RP and RAP) show that we successfully track the vast majority of large icebergs (classes A3 and A4, >10 km²) such that we can correctly identify their source. For smaller icebergs (A0-A2), that ability declines, although for A1 (0.1-1 km²) and A2 (1-10 km²) sizes, moderate performance is still achieved. This decline is to be expected since there is less geometric information available (shorter perimeters and less scope for natural shape variability) to discriminate smaller icebergs from each other while they are also more numerous, which
increases the chances of confusion. A4 was the largest class of iceberg represented in the CI2D3 dataset, but is approximately the smallest size of iceberg that would currently be named and tracked in an Antarctic context. Most named icebergs in the Antarctic are in the order $10^{10}$ m² (class A5), with the largest iceberg on record, B15, being in the order $10^{11}$ m². Consequently, our results on the CI2D3 dataset give us confidence that our method would perform well on named Antarctic icebergs as well as substantially smaller ones that are currently not routinely monitored, dramatically increasing the potential number that can

be tracked and allowing for a much more comprehensive representation of the diaspora of icebergs originating from continental sources.

We can relate, on average, over 90% of the area of icebergs back to their source when tracklets end. This implies that we are capturing the spatial distribution of most of the ice volume following large calving events (likely greater than the RAP value due to the 3-dimensional geometry of icebergs (Sulak et al., 2017)) and are able to attribute it to particular ice shelves or glaciers in situations where they calve large icebergs. This will allow us to make inferences regarding the distal impacts of changes in ice stream velocity or calving behaviour at specific locations around the coasts of either Greenland or Antarctica that may be forecast by numerical ice sheet simulations.

For operational contexts where recent motion is more informative than provenance we demonstrate a strong ability to maintain the correct identity of icebergs across multiple time intervals. The F1-score of our tracker exceeds 0.90 for lead times up to five intervals, which equates to approximately two months for the target observation frequency of the CI2D3 dataset, and remains above 0.75 to 30 intervals (approximately 1 year, (b) Figure 5). This performance provides a robust foundation for characterising iceberg motion recent to any given observation and informing inference (either human or machine-generated) about future drift patterns. Such insight represents a valuable decision support asset for navigation and hazard mitigation for fixed and mobile maritime infrastructure.

Establishing robust generational linkages is the most challenging part of the proposed tracking scheme. This is reflected in the DP, DR, and DF-1, which are lower than for the other metrics. The generational linkage procedure presented (Figure 2, Figure 3) demonstrates a clear ability to correctly align multiple child icebergs within their parent (Figure 5c) and captures a reasonable proportion of fragmentations correctly (Table 1). This is a unique capability for an automated tracking system, the performance of which will be improved upon in future work. Figure 5c(iv) also illustrates two common failure modes of generational linking .

The first failure mode is when all children are relatively small compared to the parent and a small total proportion of the parent's area is represented by its surviving children. Both such situations mean that there are few and short perimeter sections could potentially match between any one child and the parent. There is also substantial scope for a child to be incorrectly placed within the parent since the 0.96 proportional overlap heuristic can be met more easily for child icebergs that are dramatically smaller than their parent. Furthermore, the uniform vertex count when resampling polygon outlines implies that the physical vertex spacing (in metres) varies between the sub-sequences being compared for DTW distance (Figure 2) more when parent and child have dramatically different perimeter lengths. Correspondences are therefore weaker and less certain. These problems may be mitigated in future by implementing fully probabilistic matching.

The second failure mode is when there are many candidate children that are not otherwise accounted for. In Figure 5c(iv), these generational linkages are made very close to the calving front of the glacier, where many small icebergs appear near-simultaneously, but without the current method being able to represent their actual source because it is not an existing iceberg. A primary limitation of the generational matching is its greedy character that is not currently balanced by awareness of potential sources other than existing icebergs (such as calving fronts), or fates other than fragmentation (such as drifting beyond domain boundaries). This leads to erroneous linkages being made, particularly near the calving tongue of Petermann Glacier and at domain boundaries more generally. The problem could be mitigated by including the geometry of the calving tongue as a potential parent object within the tracking scheme such that newly calved icebergs could be matched to a change in calving front geometry. This would also help enhance our ability to track ice volumes right back to their sources. This was not possible in this study, using the CI2D3 dataset, because the calving front was not digitized and the underlying imagery were not available. When the fjord area was excluded (Appendix B: Exclusion of fjord) tracker performance generally improved (Table 1) which implies that incorporating calving sources could substantially improve full lineage reconstructions.

Icebergs may also appear after drifting from distal sources across the study domain boundary, while tracks may also end when icebergs drift outside the domain. In the Btrack optimization step (not used here as it is reliant upon the motion model which was disabled), hypotheses are tested that include appearance or disappearance across scene boundaries based on proximity and trajectory. Future work will implement probabilistic matching across all feasible associations based on the likelihood of geometric matches compared against the likelihood of alternative sources and fates by constructing spatial priors, like those generated for spatial filtering of potential parents (Appendix A: Probabilistic spatial filter for constraining possible parent icebergs:).

In the geospatial context of this study, the domain spans many smaller, asynchronous image volume acquisitions such that many image footprints taken at different times combine to make up the domain. The consequence of this is that at any one time where some part of the domain is observed, most of the domain is unobserved. The naïve treatment of the time domain in this study stacks observations and assigns unique timesteps to every point at which valid data are acquired anywhere in the domain. Therefore, for any given point in the domain, the temporal sequence of valid observations is sparse and non-uniform. This is the principal cause of the need for a time buffer, and for that time buffer to be relatively long (6 timesteps). As the domain gets larger, the sparsity of observations at any given location becomes more acute. This motivated the selection of a relatively small subset of the total dataset extent around the main calving fronts while retaining the majority of lineages. Nevertheless, a more sophisticated schema for handling the representation and tracking of moving objects in an asynchronously acquired domain is required if larger domains are to be studied. This problem is encountered in other domains and development of a generalized solution is beyond the scope of this study but offers an opportunity for collaboration across research disciplines.

Central to our contribution is a novel generalizable geometric assembly algorithm suited to geospatial contexts, capable of tessellating shapes to reconstruct other, larger geometries in the presence of large global invariances and imperfect correspondences between vertices. This approach should operate in any context where shapes have characteristic, high frequency, perimeter curves, although tuning of the smoothing and sliding window parameters is likely to be necessary, including when applying it to machine-generated iceberg segmentations. Applications include tracking of ice floes or reassembly of archaeological artefacts. Unlike pictorial jigsaw puzzle assembly approaches (Markaki & Panagiotakis, 2023; Shen et al., 2018), our method does not rely on any textural or image data, so is potentially more broadly applicable where only segmentation masks or silhouettes are available.

We have evaluated our approach for the CI2D3 dataset but further work is required to evaluate its generalisability to data from other sources and regions, including other areas of Greenland with differing calving regimes and for Antarctic icebergs (Guan et al., 2025). Future work will apply the approach to machine-generated segmentations and evaluate performance in an Antarctic context, then apply the tracker at a continental scale to underpin future freshwater distribution and mechanistic calving models. There is also scope for exploring supervised tracking and fragment assembly algorithms. The underlying SAR image data were not available to the authors for the purposes of this work, but if imagery corresponding to the masks in CI2D3 were available, this would offer the chance to explore supervised methods such as the transformer based cell tracking package Trackastra (Gallusser & Weigert, 2025).

## 6 Conclusions

We present a novel geospatial tracking approach for monitoring and reconstructing tracks and lineages of icebergs, evaluated against a large, unique manually annotated dataset of icebergs originating from Greenland ice tongues. We extend previous work attempting to track icebergs (Barbat et al., 2021; Koo et al., 2023; Koo et al., 2021) by developing a fully automated, unsupervised tracking methodology that establishes linkages between icebergs across fragmentation events, thus enabling reconstruction of lineage trees and full drift paths that can be traced back to the initial calving location even if the iceberg has broken up in the interim. We provide extensive evaluation of the tracker's performance using generalized metrics and those tailored to the expected downstream use cases for enhanced iceberg monitoring. This opens new opportunities to understand iceberg drift and deterioration at scale, improve iceberg motion, melt and fragmentation models as well as predict distal impacts of calving events in a much more granular manner than has hitherto been possible. The geometric assembly approach is theoretically transferrable to other domains while the whole tracking pipeline is also suited to geometry based geospatial tracking problems. The CryoTrack code (Evans, 2025) is available at https://github.com/lupinthief/CryoTrack.

**Author contributions**

BE developed the methodology and code and prepared the manuscript with contributions from all authors. AL developed tracking functionality for Btrack to enable this work and advised on approach. AC assisted in data provision, design and manuscript preparation. AF and SH conceived the work and assisted in methodological design and manuscript preparation.

**Code and data availability**

The CryoTrack code is available at https://github.com/lupinthief/CryoTrack. The CI2D3 database is available at https://www.polardata.ca/pdcsearch/PDCSearch.jsp?doi_id=12678

**Competing interests**

The authors declare that they have no conflict of interest

**Acknowledgements**

This work was funded by EPSRC Grant EP/Y028880/1, The Alan Turing Institute, and the Polar Science for a Sustainable Planet programme at the British Antarctic Survey. We are grateful for productive discussions with, among others, Martin Rogers, Louisa Van Zeeland, Ellen Bowler, Arianna Salili-James, Cameron Trotter, Andreas Bock, James Byrne and Jonathan Smith. GitHub Copilot was used during code development.

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

**Appendix A: Probabilistic spatial filter for constraining possible parent icebergs:**

The search domain for potential parent icebergs when conducting the generational linkage stage is constrained by vector fields learned from the tracklets generated for unchanged iceberg identities and the time-lag between observations of the child and potential parent icebergs.

Tracklets are initially temporally densified such that each arc represents a single time step. This is achieved by linear interpolation of the iceberg locations between start and end point for cases where an arc's duration is greater than one time
interval. Radial Basis Function interpolation (scipy.interpolation.RBFInterpolator ((Virtanen et al., 2020), linear kernel, smoothing 1e5) is then applied to the tracklet arcs with uniform time duration (1) and predicted onto a 25 by 25 grid covering the study domain to produce vector fields describing the interpolated motion of icebergs dependent on their location within the domain (vx and vy). These are shown in Figure A1.

When constraining potential parents for a child iceberg, probabilistic fields of source location are generated by 'backtracking'
through the vector fields for the number of time intervals between the child observation and the potential parent observation,
starting at the grid centroid closest to the child observation. At each time interval, the source probabilities for the location are
calculated based on the vector fields and a gaussian representation of uncertainty (we used σ=5000 as a compromise between
the standard deviations within our observed vector field ($\sigma_{vx}$ = 3392 m, $\sigma_{vy}$ = 8662 m)) and accumulated over the number of
timesteps before being normalised in 0-1. The result is a probability field describing likelihoods for the source location of the
child iceberg at a given lead time (inset panels to Figure A1). If a potential parent is located such that its associated probability
of being a source is above a given, tuned, threshold of 0.05 (e.g it falls within the contour in Figure A1) it is included in the
list of potential parents for that child iceberg.

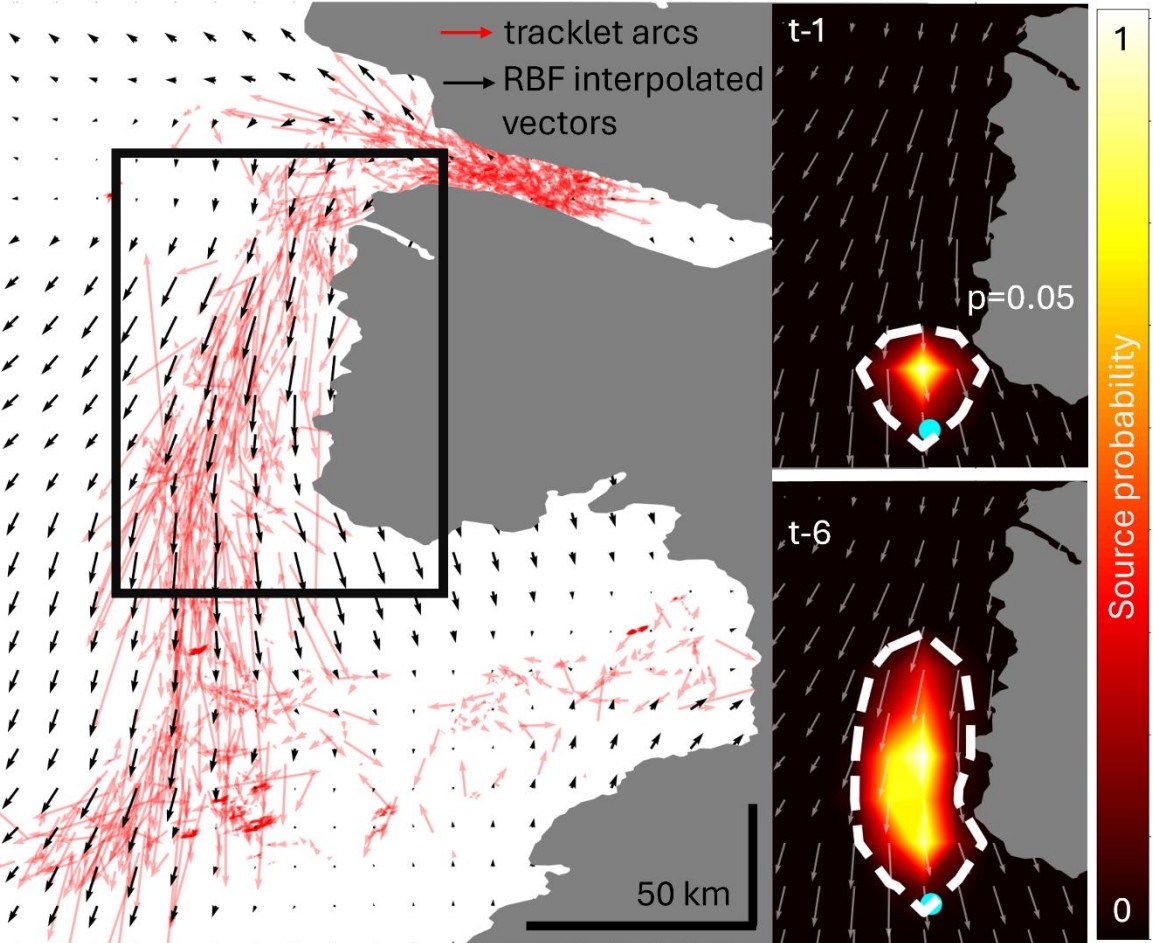

**Figure A1 - temporally-densified tracklet arcs (red arrows) and radial basis function interpolated vector field (black arrows). Insets
show illustrative probabilistic source map for example location of cyan dot within sub-region (black box) for lead times of 1 (top
tight) and 6 (bottom right), with p=0.05 contour shown.**

**Appendix B: Exclusion of fjord**

We tested the performance of generational linking in locations outside the fjord of Petermann Glacier. Within the fjord there is a propensity of the method to allocate newly-appearing icebergs to fragmentation of existing icebergs when in reality they are calving from the glacier tongue. As outlined in the discussion, this arises because out dataset does not include digitisations of the shape of the calving front itself so the tessellation process cannot allocate new tracklets to it as a source. Where the process finds a potential generational linkage, therefore, it allocates it without comparing with any geometric fit to the calving front.

To assess the impact of this limitation on the performance of our generational linking we evaluated our tracks against a subset of the dataset that excludes the fjord. Figure B1 shows the fjord area with only those icebergs calved in 2008 shown for clarity. All iceberg outlines that intersected with an area representing the fjord (hatched on  Figure B1) were excluded from the dataset. The orange filled iceberg re-entered the fjord after being observed in this location and subsequently fragmented, with its children first being observed outside the fjord. The ID of this iceberg was manually updated to that of its last observed instance

within the fjord prior to fragmentation to allow for correct assessment of the lineage of its children. The remainder of the tracking and evaluation procedure was unchanged.

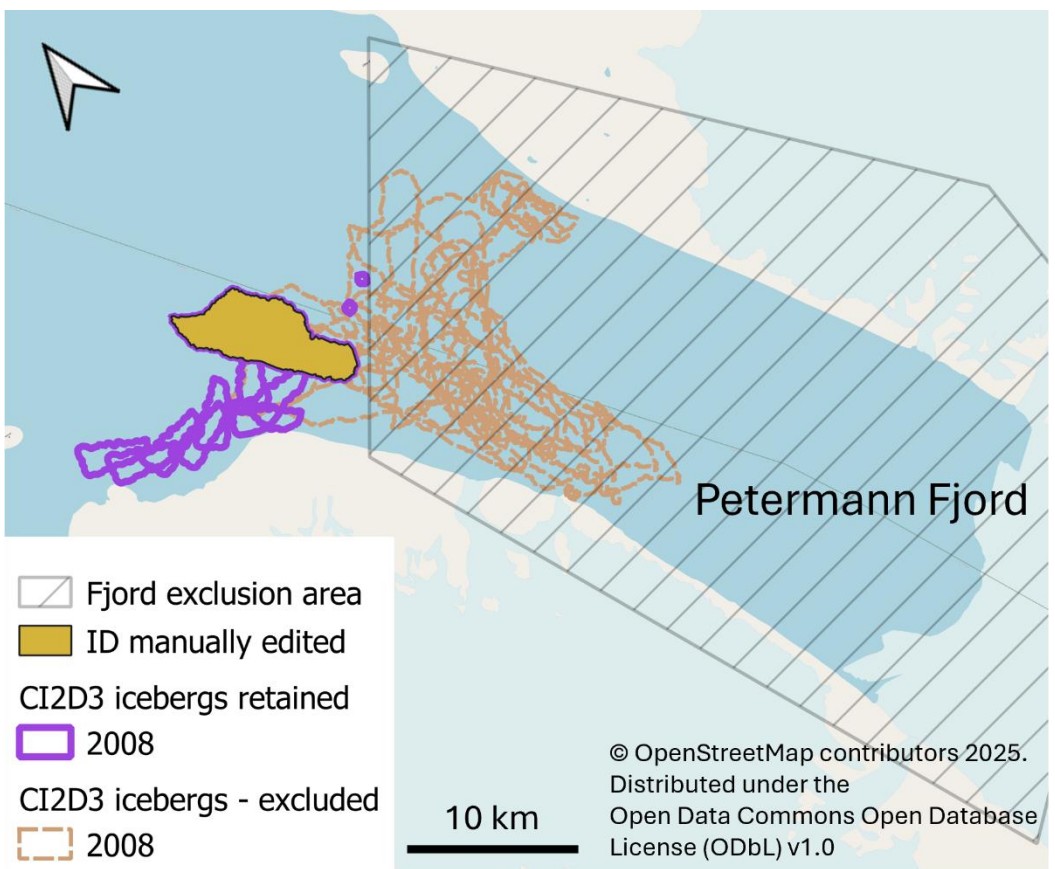

**Figure B1 - Exclusion of icebergs within Petermann Fjord, showing only 2008 icebergs for clarity. Those intersecting with hatched fjord area were removed from dataset. The filled iceberg re-entered the hatched area before fragmenting so its ID was updated manually to allow correct evaluation of the lineage of the fragments. Map data: https://www.openstreetmap.org/copyright.**