# Peer review of "Icebergs, jigsaw puzzles and genealogy: Automated multi-generational iceberg tracking and lineage reconstruction."

_EGUsphere, 2025_

## Author Response (AR1)

`Response to Anonymous Reviewer #1:

**We thank the reviewer for their detailed reading of our manuscript, appreciation of the value of tackling the problem of iceberg tracking and for the insights offered. We have done our best to address the questions and concerns raised and provide responses and detail on the revisions made to the manuscript in bold line with their suggestions below.**

This study proposes a very interesting iceberg tracking methodology, which is based on live cell tracking but modified for the geometric context of icebergs. Although their approach is only tested within a small area covered by the CI2D3 database around the Petermann Glacier in northern Greenland, their method showed a significant performance in tracking iceberg trajectories and linking parent and child icebergs. I value the authors' novel idea of combining and adopting cell tracking techniques for iceberg tracking, and this study can particularly contribute to efficient monitoring of the formation, movement, and fragmentation of icebergs. However, the authors have to provide (1) a more detailed description of their method and (2) sources of uncertainties in their tessellation, probably with some representative examples. Please see my detailed comments below.

**Thank you for your recognition of the value of the work. We hope that the detail below and changes to the manuscript fully address your two concerns listed above.**

- L152: Why is this 256-vertex resampling necessary? How did you determine this specific number of vertices? Does this mean all icebergs are resampled into 256 vertices regardless of their sizes? Shouldn't a large iceberg have more vertices because its shape is more complex?

**The issue of whether to resample and if so, in what manner, was raised by both reviewers and poses an interesting question that we feel requires a judgement call as to the relative merits of resampling or not resampling.**

**We feel that some degree of resampling is imperative to allow for transferability of the downstream components of the processing between iceberg outlines generated by different vectorisation methods without risking excessive computational overheads. For example, we work in this study with manually delineated outlines generated by clicking to produce successive vertices in the shape. This produces outlines with relatively sparse vertices, typically in the order of a few hundred even for the larger icebergs. In contrast, a machine-generated segmentation or one produced manually by a continuous tracing method would produce outlines potentially containing tens or hundreds of thousands of vertices as each pixel boundary would be traced. Such large numbers of vertices would prove computationally prohibitive when calculating the DTW distance matrix (Figure 2c) for all pairs of perimeter subsets. Furthermore, using the original polygon vertices does not, in itself, imply that the real-world vertex spacing is consistent, since this depends on the vectorisation method used, so resampling does not imply losing this information.**

**Having decided that some resampling was necessary for generalisability we were faced with decisions over how to resample the outlines. Options were to resample to a consistent number of vertices or to resample to a consistent real-world vertex spacing. We opted to resample to a consistent number of equally-spaced vertices for two reasons. Firstly, we suspect that the geometries of iceberg outlines exhibit a degree of fractal behaviour – that is, they are likely to be self-similar irrespective of the scale at which they are observed. We recognise that this is a hypothesis and evidence is sparse due to a lack of work in this domain. Nevertheless, if there is a degree of fractal behaviour in perimeter**

geometries and we resample the outline to a consistent physical vertex spacing then we are effectively observing the outlines of each iceberg at the same scale and may expect them to exhibit strong similarities, potentially making them harder to distinguish between. If, however, we resample to a fixed number of vertices, then we sample the larger iceberg outlines more sparsely. When we come to take the Euclidean distances between these and the EFD representation of the outline (figure 2a), the physical dimensions (in metres) are retained. Therefore, the absolute magnitude of the deviations for a more coarsely sampled, larger outline tends to be greater than for a densely-sampled, small outline. In this way we are able to propagate some scale-awareness to the DTW distance calculation in the amplitude domain, even though our resampling approach loses some information in the frequency domain. This helps to favour matches of objects that are more similar in size to each other over objects of substantially different sizes. Furthermore, the EFD reconstruction of the outline against which distances are calculated also requires a number of vertices to be determined, allowing better vertex-to-vertex correspondences when calculating minimum Euclidean distances. We recognise that the alternative approach of resampling to a uniform physical spacing could bring advantages when matching small icebergs to much larger ones as it would potentially improve the reliability with which small-amplitude deviations are matched but we consider the priority to be to successfully match the larger icebergs since these represent the greatest sources of freshwater and are the most important to understand the provenance of. We consider that the uniform number resampling supports this priority better than the uniform distance sampling. The DTW distance matrix calculation is one of the most computationally expensive parts of the workflow so resampling to a uniform number of vertices is also more computationally tractable, since the perimeter of the large icebergs can be many times that of the small ones and defining a uniform spacing that provides sufficient vertices to describe the small icebergs would imply thousands or tens of thousands of vertices for the largest icebergs at the same spacing, dramatically increasing computational costs. This would be further exacerbated in the context of Antarctic icebergs that can be orders of magnitude larger than the largest within the CI2D3 dataset used here.

The number of vertices was selected pragmatically so as not to dramatically over-sample the smaller, simpler outlines relative to their outline detail in the manual vectorisations and to minimise undue computational overheads while still providing sufficient vertices to describe the larger-scale complexity of the larger icebergs.

We have added the following text to the manuscript to briefly outline this reasoning and hope that, in conjunction with this more detailed, public explanation, this suffices to explain the design choice we made:

*"The 256 vertex resampling ensures that, even for very large icebergs, the outline alignment stage (Error! Reference source not found.) remains computationally tractable, which would not be guaranteed if using a uniform-distance resampling or without resampling at all. Furthermore, resampling to a uniform number of vertices helps to propagate some scale awareness to the amplitude component of the 1-d distance vectors (Error! Reference source not found.b) upon which iceberg associations are based, helping to exploit information on the relative sizes of the iceberg when proposing matches." (L166-171, revised MS)*

- L167: coeficients -> coefficients

**Changed**

- L180: btrack -> Btrack

**Changed**

- L174-178: So, does this mean that an iceberg object with the minimum cosine distance from the previous timestep is determined as the same identity iceberg, but if there is no object with a cosine distance below a certain threshold (e.g., 0.05), the tracklet for this iceberg is not generated? The description of iceberg tracking here is not clear, so please clarify this process in detail.

**Thank you for making us aware that this stage is unclear. We have updated the manuscript to include a clearer explanation while remaining brief, and to point the reader to a detailed explanation in** Ulicna et al. (2021)**, should they want further detail. We have also made it explicit that the configuration file for this stage is available alongside the codebase, ensuring reproducibility:**

*"We then use Bayesian Tracker (Btrack, Ulicna et al. (2021)), a python package developed for live cell tracking, to establish tracklets for which geometric characteristics do not change dramatically. We use the 'visual features' linking but disable the motion model that places spatial priors on future iceberg locations since it is poorly suited to predicting the highly variable movement of icebergs and the non-uniform time spacing of observations. We also do not conduct global optimization, the step in which Btrack attempts to construct links between tracklets and establish parent-child relations since the heuristics are not appropriate for the iceberg context (see introduction). In the process of tracklet generation, Btrack constructs a Bayesian belief matrix for each timestep with uniform prior and dimensions N x (M+1), where N is the number of existing tracks and M is the number of objects detected in the current field of view. Bayesian updates are then performed based on cosine distances between the feature vectors for all pairs of icebergs within a given search radius of each other to calculate the probability of a link being established or the object being considered lost (by reference to a tunable parameter). Finally iceberg associations are chosen, given the belief matrix, based on the maximum posterior probability of either an association or loss of the tracklet. Icebergs in the current frame that have not been associated with an existing tracklet generate a new tracklet while lost tracklets persist as dummies for a prescribed number of timesteps (see below). Using the five visual features, the median cosine distance between icebergs and other temporal instances of the same identity was $3.2E^{-9}$, whereas the median distance to the icebergs with a different identity was seven orders of magnitude larger at 0.05. This indicates effective separation of geometries in this 5-dimensional feature space. To handle the temporal data sparsity problem arising from the large domain and intermittent satellite coverage of any one location within it, Btrack is able to insert dummy instances for a prescribed number of timesteps between linked observations. If an iceberg is not observed again within the given time buffer the tracklet is terminated. The search radius and time buffer are tunable parameters that were set, through experimentation, at 100km and 6 timesteps respectively. Optimal values of these will be a function of the domain extent, data frequency and environmental factors controlling iceberg motion. Increasing them will tend to increase the false positive linkage rate while decreasing them will tend to*

***increase the false negative rate.*** *Ulicna et al. (2021)* ***provide a detailed explanation of how Btrack constructs tracklets, and the reader is referred there for further detail. The configuration file for the Btrack step is available alongside the codebase (see code availability)."*** (**L186-210, revised MS**)

- L207-208: According to Figure 2b, it seems that this 1-D vector of deviations is a function of vertex number (256 vertices). If that is the case, I believe this 1-D vector would be highly affected by how the 256 vertices are resampled. That is, even identical icebergs can have different 1-D vector shapes depending on the way these 256 vertices are resampled. Therefore, I encourage the authors to add a detailed description of how they conduct the 256-vertex resampling.

**We agree that there is the potential for different representations based on resampling strategy, although we do not feel that the overall process would be particularly sensitive to these beyond the effects discussed above, assuming a consistent strategy is applied. We have added detail upon first mention of the 256 point resampling to make explicit that the interpolation is equal-spacing and point the reader to the public codebase where they can see the detail of the implementation:**

**"Each polygon in the CI2D3 Database is represented by its geometry, which we resampled to a uniform 256 vertices equally spaced around the perimeter (see codebase for implementation)" (L163-164, revised MS)**

- L308: Here, the authors mentioned that they varied the effective search radius, but they also mentioned that the search radius is set to 100 km in L181-182. Are these two search radii different? Or does the 100 km search radius come from these sigma and probability threshold parameters? Please clarify this.

**We realise that we didn't make this process explicit outside the supplementary material in Appendix A. We have added the following to the main text to clarify the distinction between the 100km search radius used in tracklet generation and the variable domain used in generational linkage:**

**"Starting with the largest candidate child, we identify possible parents within the preceding time range using a probabilistic spatial filter (process 3,** Error! Reference source not found.**) based upon vector fields interpolated from the tracklet data (see** Error! Reference source not found.**). In contrast to the fixed search radius of 100km used for tracklet construction, this allows us to inform where we look for matches based upon the tracklet observations and the time interval between observations, helping to constrain the locations of proposed generational linkages to be consistent with the observed motion of icebergs between fragmentation events." (L273-279, revised MS)**

- Figure 4: This 3D figure looks cool, but the 3D locations of tracks in the x, y, and t axes are hard to interpret. It would be better to include this as a 3D animation in supplementary materials; instead, the authors can just provide 2D maps of iceberg locations and shapes for multiple time steps on multiple panels.

**Thank you for this suggestion. We have updated figure 4 with 2-D maps, along with its caption and description in the text. We have used colours to denote whether arcs and generational linkages are TP, FP or FN. We hope the revised figure and description provides more insight and clarity.**

**"Figure 1 shows examples of lineages reconstructed using our method are shown for two timepoints (t=341 in main panel and t=370 in inset) to illustrate correct tracks and various possible failure modes. Five points of interest (A-E) are marked. A shows a fragmentation event that is identified by white circles on both panels to aid orientation. This event produced two FP generational linkages that also imply two FN arcs. B marks a single FN arc in an otherwise long and correct track for a small iceberg. C marks the fragmentation shown in more detail in** Error! Reference source not found.**c(iii) where three children are correctly matched and one missed. D marks a correctly tracked fragmentation into two children and E shows successive failures (both FP and FN) in the track of a very small iceberg"** (L361-367, revised MS)

"Figure 1: Example tracks reconstructed from the CI2D3 dataset prior to two timepoints (t=341 in main panel, t=370 for inset). True positive (TP), false positive (FP) and false negative (FN) arcs and generational links are shown. Point A (white circle) denotes a fragmentation resulting in one TP generational link and two FP generational links. This event is identified in both pan els for orientation purposes. B denotes a FN link in an otherwise long correct track, C corresponds to the fragmentation shown in Error! Reference source not found.c(iii), D marks a correctly tracked fragmentation, E denotes a very small, poorly tracked iceberg with both FP an FN arcs Greenland elevation data from GIMP-DEM 90 (Howat et al., 2014)." (caption, Figure 4)

- L317: Does the 0.96 proportional overlap indicate the 96 % overlap threshold specified in L219-220? If so, please clarify this in the text. I would also like to know the units of sigma, time buffer, and sub-section length.

**Yes, the 0.96 corresponds to 96% overlap threshold. We have adapted the text to make this clear and provide units:**

***"The final configuration for which we report performance used the following parameters: sigma = 5000 m; probability threshold = 0.05; proportional overlap = 0.96 (corresponding to the 96% threshold described above); time buffer = 6 timesteps; sub-section length = 10 vertices"*** *(L359-361, revised MS)*

- Figure 5b: Does every lead time (in the number of timesteps) correspond to any real time interval (e.g., hours or days)? If so (I believe so based on the description in L146-147 and L383-384), it would be better to replace this lead time with a physical time interval in hours or days. If not (i.e., lead time is not constant), please specify it in the caption or text.

**No, the timesteps are irregular due to the asynchronous acquisition of images of different parts of the domain by the satellites. It doesn't, therefore, make sense in this context to convert to physical time intervals. We treat this problem in some detail in both the methods (L153-163, revised MS) and the discussion and recognise that innovations are needed to overcome it in future work (L480-491, revised MS), but have adjusted the text and caption to make the non-uniform time interval more explicit:**

**"For the purposes of demonstrating the proposed method, the dates at which any observation was contained in the database were stacked and a uniformly incrementing timestep assigned to that date, implying that the physical time interval between successive timesteps is non-uniform." (L158-160, revised MS)**

**"(b) Performance in maintaining iceberg identity over 50 non-uniform time intervals."
(caption, fig. 5)**

**"Therefore, for any given point in the domain, the temporal sequence of valid observations
is sparse and non-uniform" (L484, revised MS)**

- Figure 5c: Although the authors specify that these figures are not to scale, I encourage the
authors to at least add a scale bar for each iceberg so readers can have insight into the RP
performance by iceberg size.

**We have added more examples and included scale bars for each panel**

- L349-353: Figure 3c shows successful examples of iceberg generational associations.
However, it would also be important to understand why this method fails in many other cases;
DP, DR, and DF-1 look pretty low in Table 1, even when excluding fjord from the domain. I
encourage the authors to add several examples of failures in Figure 3 and add comprehensive
discussions about the reasons for these failures.

**We believe the reviewer is referring to Figure 5c, which we have updated to include
additional examples, including true positive, false positive and false negative matches for
a range of iceberg sizes. We have updated the figure caption and description in the results
section accordingly:**

**"(c) examples of automated tessellations (arbitrary scale and colours with contrasting
outlines to illustrate where fitted shapes overlap. True positive (green plus) and false
positive (red minus) associations are indicated. Shapes without outlines that fall outside
the red parent outlines for (iii) and (iv) are false negative associations." (caption Fig 5)**

**"Examples of tessellations enabling reconstruction of complex many-to-one generational
associations are shown in** Error! Reference source not found.**c illustrating various success
and failure modes. Panels i and ii show wholly correct tessellations for medium and small
parent icebergs respectively. Panel iii shows a large parent iceberg with three correct child
linkages but one false negative association (shown outside parent outline) that was not
made. Panel iv shows a largely incorrect set of linkages with only one child correctly
attributed and four false positives and four false negatives." (L401-405, revised MS)**

**We have also added a detailed discussion of the failure modes illustrated and how they
arise:**

**"**Error! Reference source not found.**c(iv) also illustrates two common failure modes of
generational linking .**

**The first failure mode is when all children are relatively small compared to the parent and a
small total proportion of the parent's area is represented by its surviving children. Both
such situations mean that there are few and short perimeter sections could potentially
match between any one child and the parent. There is also substantial scope for a child to
be incorrectly placed within the parent since the 0.96 proportional overlap heuristic can be
met more easily for child icebergs that are dramatically smaller than their parent.
Furthermore, the uniform vertex count when resampling polygon outlines implies that the
physical vertex spacing (in metres) varies between the sub-sequences being compared for
DTW distance (**Error! Reference source not found.**) more when parent and child have
dramatically different perimeter lengths. Correspondences are therefore weaker and less**

**certain. These problems may be mitigated in future by implementing fully probabilistic matching.**

**The second failure mode is when there are many candidate children that are not otherwise accounted for. In** Error! Reference source not found.**c(iv), these generational linkages are made very close to the calving front of the glacier, where many small icebergs appear near-simultaneously, but without the current method being able to represent their actual source because it is not an existing iceberg. "** (L447-461, revised MS)

- L363-368: Can the authors add an Appendix (or directly add to this section or result section) to show several examples of failure in identifying iceberg origins (i.e., low RP) in terms of iceberg size? It would be good to see the effect of iceberg size through real examples.

**We hope that the revised Figure 4 and accompanying description, coupled with the changes made to Figure 5c and its discussion (see other responses) help to better exemplify contexts where the tracker is accurate and those where it fails.**

- L404: btrack -> Btrack

**Changed**

- L429-434: I notice that an interesting research paper about the Antarctic iceberg fragmentation database has been recently published: Guan et al., 2025. I'm not sure if the authors of this publication have shared their data with the public, but it would be interesting to assess this new technique with this Antarctic iceberg fragmentation database as well in the future.

**Thank you for suggesting that we include this work. We have added a reference to it as a valuable resource for future testing and development of the proposed method:**

**"We have evaluated our approach for the CI2D3 dataset. Further work is required to evaluate its generalisability to data from other sources and regions, including other areas of Greenland with differing calving regimes and for Antarctic icebergs** (Guan et al., 2025)**."** (L502-504, revised MS)

Guan, Z., Liu, Y., Cheng, X., Li, T., Shokr, M., Liu, X., ... Chen, Z. (2025). Fragmentation patterns of Antarctic icebergs in sea ice: observations and statistical data. *International Journal of Digital Earth*, *18*(1). https://doi.org/10.1080/17538947.2025.2511289

**Once again, we are grateful to the reviewer for their insight and assistance in improving our manuscript. We hope that we have fully addressed all their questions.**

Response to Anonymous Reviewer #2:

**We thank the reviewer for their detailed reading of our manuscript, appreciation of the value of tackling the problem of iceberg tracking and for the insights offered. We have done our best to address the questions and concerns raised and provide responses and detail on the revisions made to the manuscript in bold in line with their suggestions below.**

Review "Icebergs, jigsaw puzzles and genealogy: Automated multi-generational iceberg tracking and lineage reconstruction"

The study presents an innovative approach for tracking icebergs, including their fragmentation. Iceberg tracking remains a challenging task, still requiring substantial manual work, while it is essential for understanding a wide range of processes. The proposed methodology shows promising results with the potential to overcome some of the main challenges in iceberg tracking. Although the approach has so far been tested on a relatively small dataset, it shows promising results in automated monitoring of iceberg formation, movement, and fragmentation. However, the study would benefit from a more detailed description of the new method, including examples showing real data with more than one iceberg, as well as a clearer discussion of its limitations and how they may affect the results. Additionally, the method was tested in the vicinity of Petermann Glacier. While it appears to be transferable, this claim should be tested by applying it to other regions (also for the large amount of glaciers in Greenland calving very frequent small sized icebergs). I agree that this is out of the scope for this manuscript but it should be clearly stated.

**We thank the reviewer for their positive assessment of our work. We are unsure, however, as to what they are asking for in reference to examples showing real data with more than one iceberg. Across figures 1, 2, 3 and 5 we already show examples from five different parent-child associations, and have added an extra two to figure 5 showing false positive associations. The revised Figure 4 also contains multiple icebergs and lineages at multiple timesteps. We therefore feel that we have provided plenty of visual examples. We recognise, however that the description of limitations and effects on results can be improved and have endeavoured to do so in line with the suggestions below. We appreciate that the reviewer recognises that testing on other regions is necessary but out of scope because of a lack of alternative 'ground truth' datasets documenting lineages.**

Abstract:

I would appreciate it if it became clearer already in the abstract that the approach is not tackling the problem of iceberg detection, which remains a challenge, inducing a lot of uncertainty, which indirectly effects the tracking and its validation.

**We have adapted the abstract text to make clear that the current study does not address iceberg detection.**

**"This system, which focuses on the tracking of icebergs, but not the related and challenging problem of their detection, contributes to the need for scalable iceberg monitoring." (L24-25, revised MS)**

L10: The first automated multi-generational approach, probably not the first approach overall.

**We have added 'multi-generational' to clarify:**

**"This study presents the first comprehensively validated, scalable multi-generational iceberg tracking approach..." (L10-13, revised MS)**

Introduction:

L31-32: There is newer literature available.

**We have added references to the work of Mottram et al.,2024, Coulon et al, 2024 and Davison et al., 2020 (L31-32, revised MS)**

Data, Methods & Data preparation:

What is the temporal resolution of the different data sources?

**We already address the temporal resolution and its implications in detail in the methods and discussion (L153-163 and L480-491, revised MS), but have added a brief detail here to clarify early-on:**

**"manually delineated from a combination RADARSAT-1 and -2 and Envisat imagery selected with a target revisit period of two weeks..." (L108-109, revised MS)**

All three chapters need a more detailed description for better understanding.

**We hope that the additional detail provided under the reviewer's specific points below serves to improve the descriptions.**

The limitations of the dataset influence the tracking and its validation and should be discussed already here.

**We agree, and have added an example to acknowledge the potential impacts on our results:**

**"Nevertheless, the reference dataset's limitations will affect the tracking results. For example, we have observed that at least one iceberg with near identical geometry and close proximity that we believe to be the same iceberg, but which lacks a track linking the observations in the CI2D3 dataset. Such artefacts of the manual annotation process are believed to be rare but have the potential to affect the performance metrics for our automated tracking approach." (L113-117, revised MS)**

It is not completely clear to me how the fragmentation was handled in the Database used for tracking and validation? Was it a manual decision which icebergs are the children of which parental iceberg? Does this not imply severe uncertainties?

**Yes, generational associations are manually assigned, but human interpretation is currently the only viable means of generating such data for so many icebergs, and this is precisely the problem that our work seeks to help overcome. We have specified the process and acknowledged that it has uncertainties in the text:**

**"Lineage associations were manually ascribed by the expert annotator, taking into account proximity, shape and appearance including surface patterns and textures. While manual**

**determination of lineages implies a degree of uncertainty, it represents the most reliable method available." (L111-113, revised MS)**

L148: How do you stack the observations? This step is not clear to me. Does this mean an iceberg considered to be at time x at position y is in reality at time z at position y? I agree that for the development of the approach this is not relevant but it is a clear limitation of the approach, if it is not able to handle non-uniform time steps, which should become clearer from the descriptions. How does the algorithm deal with data gaps?

**Successive observations are stacked sequentially by date of image, but images do not necessarily cover the entire study domain meaning that the source data are spatiotemporally sparse. We already treat this problem, the time buffer used to handle it alongside the implications thereof and the need for a more sophisticated solution to overcome it for future work in some detail in the methods (L153-163, revised MS) and discussion (L480-491, revised MS) sections. We have added detail here to ensure that it is clear that the time interval is non-uniform but leave the discussion of the implications for other sections:**

**"For the purposes of demonstrating the proposed method, the dates at which any observation was contained in the database were stacked and a uniformly incrementing timestep assigned to that date, implying that the physical time interval between successive timesteps is non-uniform" (L157-159, revised MS)**

L152: Why 256 vertices? Why not adapting the number of vertices depending on the size of the icebergs e.g. uniform length of the vertices?

**The issue of whether to resample and if so, in what manner, was raised by both reviewers and poses an interesting question that we feel requires a judgement call as to the relative merits of resampling or not resampling.**

**We feel that some degree of resampling is imperative to allow for transferability of the downstream components of the processing between iceberg outlines generated by different vectorisation methods without risking excessive computational overheads. For example, we work in this study with manually delineated outlines generated by clicking to produce successive vertices in the shape. This produces outlines with relatively sparse vertices, typically in the order of a few hundred even for the larger icebergs. In contrast, a machine-generated segmentation or one produced manually by a continuous tracing method would produce outlines potentially containing tens or hundreds of thousands of vertices as each pixel boundary would be traced. Such large numbers of vertices would prove computationally prohibitive when calculating the DTW distance matrix (Figure 2c) for all pairs of perimeter subsets. Furthermore, using the original polygon vertices does not, in itself, imply that the real-world vertex spacing is consistent, since this depends on the vectorisation method used, so resampling does not imply losing this information.**

**Having decided that some resampling was necessary for generalisability we were faced with decisions over how to resample the outlines. Options were to resample to a consistent number of vertices or to resample to a consistent real-world vertex spacing. We opted to resample to a consistent number of equally-spaced vertices for two reasons. Firstly, we suspect that the geometries of iceberg outlines exhibit a degree of fractal behaviour – that is, they are likely to be self-similar irrespective of the scale at which they are observed. We recognise that this is a hypothesis and evidence is sparse due to a lack**

of work in this domain. Nevertheless, if there is a degree of fractal behaviour in perimeter geometries and we resample the outline to a consistent physical vertex spacing then we are effectively observing the outlines of each iceberg at the same scale and may expect them to exhibit strong similarities, potentially making them harder to distinguish between. If, however, we resample to a fixed number of vertices, then we sample the larger iceberg outlines more sparsely. When we come to take the Euclidean distances between these and the EFD representation of the outline (figure 2a), the physical dimensions (in metres) are retained. Therefore, the absolute magnitude of the deviations for a more coarsely sampled, larger outline tends to be greater than for a densely-sampled, small outline. In this way we are able to propagate some scale-awareness to the DTW distance calculation in the amplitude domain, even though our resampling approach loses some information in the frequency domain. This helps to favour matches of objects that are more similar in size to each other over objects of substantially different sizes. Furthermore, the EFD reconstruction of the outline against which distances are calculated also requires a number of vertices to be determined, allowing better vertex-to-vertex correspondences when calculating minimum Euclidean distances. We recognise that the alternative approach of resampling to a uniform physical spacing could bring advantages when matching small icebergs to much larger ones as it would potentially improve the reliability with which small-amplitude deviations are matched but we consider the priority to be to successfully match the larger icebergs since these represent the greatest sources of freshwater and are the most important to understand the provenance of. We consider that the uniform number resampling supports this priority better than the uniform distance sampling. The DTW distance matrix calculation is one of the most computationally expensive parts of the workflow so resampling to a uniform number of vertices is also more computationally tractable, since the perimeter of the large icebergs can be many times that of the small ones and defining a uniform spacing that provides sufficient vertices to describe the small icebergs would imply thousands or tens of thousands of vertices for the largest icebergs at the same spacing, dramatically increasing computational costs. This would be further exacerbated in the context of Antarctic icebergs that can be orders of magnitude larger than the largest within the CI2D3 dataset used here.

The number of vertices was selected pragmatically so as not to dramatically over-sample the smaller, simpler outlines relative to their outline detail in the manual vectorisations and to minimise undue computational overheads while still providing sufficient vertices to describe the larger-scale complexity of the larger icebergs.

We have added the following text to the manuscript to briefly outline this reasoning and hope that, in conjunction with this more detailed, public explanation, this suffices to explain the design choice we made:

*"The 256 vertex resampling ensures that, even for very large icebergs, the outline alignment stage (Error! Reference source not found.) remains computationally tractable, which would not be guaranteed if using a uniform-distance resampling or without resampling at all. Furthermore, resampling to a uniform number of vertices helps to propagate some scale awareness to the amplitude component of the 1-d distance vectors (Error! Reference source not found.b) upon which iceberg associations are based, helping to exploit information on the relative sizes of the iceberg when proposing matches."* **(L165-170, revised MS)**

Tracklet construction:

L170: What means do not change dramatically? Are there thresholds?

**No, there are not thresholds. This depends on the posterior likelihood of a tracklet being generated during Bayesian updates used internally by Btrack. We have updated the text extensively to provide more detail on how Btrack operates and to point to a more detailed explanation:**

**"We then use Bayesian Tracker (Btrack, Ulicna et al. (2021)), a python package developed for live cell tracking, to establish tracklets for which geometric characteristics do not change dramatically (i.e. they are similar enough that Btrack can recognise them as the same iceberg across successive observations). We use the 'visual features' linking but disable the motion model that places spatial priors on future iceberg locations since it is poorly suited to predicting the highly variable movement of icebergs and the non-uniform time spacing of observations. We also do not conduct global optimization, the step in which Btrack attempts to construct links between tracklets and establish parent-child relations since the heuristics are not appropriate for the iceberg context (see introduction). In the process of tracklet generation, Btrack constructs a Bayesian belief matrix for each timestep with uniform prior and dimensions N x (M+1), where N is the number of existing tracks and M is the number of objects detected in the current field of view. Bayesian updates are then performed based on cosine distances between the feature vectors for all pairs of icebergs within a given search radius of each other to calculate the probability of a link being established or the object being considered lost (by reference to a tuneable parameter, see config file). Finally, iceberg associations are chosen, given the belief matrix, based on the maximum posterior probability of either an association or loss of the tracklet. Icebergs in the current frame that have not been associated with an existing tracklet generate a new tracklet while lost tracklets persist as dummies for a prescribed number of timesteps (see below). Using the five visual features, the median cosine distance between icebergs and other temporal instances of the same identity was $3.2E^{-9}$, whereas the median distance to the icebergs with a different identity was seven orders of magnitude larger at 0.05. This indicates effective separation of geometries in this 5-dimensional feature space. To handle the temporal data sparsity problem arising from the large domain and intermittent satellite coverage of any one location within it, Btrack is able to insert dummy instances for a prescribed number of timesteps between linked observations. If an iceberg is not observed again within the given time buffer the tracklet is terminated. The search radius and time buffer are tunable parameters that were set, through experimentation, at 100km and 6 timesteps respectively. Optimal values of these will be a function of the domain extent, data frequency and environmental factors controlling iceberg motion. Increasing them will tend to increase the false positive linkage rate while decreasing them will tend to increase the false negative rate. Ulicna et al. (2021) provide a detailed explanation of how Btrack constructs tracklets, and the reader is referred there for further detail. The configuration file for the Btrack step is available alongside the codebase (see code availability)." (L185-209, revised MS)**

Use consistently Btrack throughout the manuscript.

**We have changed all instances to 'Btrack'**

Generation linking:

What happens in your approach if two icebergs get very close and might be identified as one at a timestep?

**This confusion would be an artefact of an imperfect segmentation process used to generate the polygon outlines, so we do not address it directly here. As long as polygons are closed and have a unique ID, their proximity will not result in confusion of this sort. In the process of generating the polygons, which is outside the scope of the current work, it is possible that touching icebergs would become a single polygon. In this case, two previously observed icebergs may appear lost when they touch and a new iceberg appear. If they then separate again within the time buffer and become visible as two distinct icebergs that are similar in shape to before they touched, then the tracklet generation stage using Btrack will probably see them as continuations of their original tracklets. If they do not separate within that time period they will generate a new tracklet representing the combined outline, that will persist until they separate, at which point this will be treated as a fragmentation event.**

While being a large step forward, this step also poses uncertainties and limitations. It would be good if the authors could elaborate more on this.

**We welcome the acknowledgement of the progress this represents and fully agree that there are limitations and uncertainties associated. We mention in the discussion that in future work we plan to implement fully probabilistic matching, which would allow for a better description of the confidence of associations and the uncertainties within lineages. There is an extreme paucity of observational studies evaluating generational linkages, which arises from the difficulty of the challenge and the labour-intensive methods currently available to address it. We feel that even our imperfect, yet scalable method will offer scope for substantial new insights. We already treat the limitations of this method and possible avenues for improvement in considerable detail in the discussion, but have edited this section to point the reader to the later discussion:**

**"We discuss the limitations and uncertainties that arise, along with further work required to improve the performance of this step below. (L303-304, revised MS)**

**How the uncertainties propagate into downstream tasks will depend very much upon what those tasks are, so we cannot pre-suppose the resulting limitations for all possible use-cases, and therefore feel that further supposition would not be helpful within this manuscript.**

Outline alignment:

In this chapter are a lot of abbreviations and it would help if they were explained the first time they are used.

**We have carefully checked that all abbreviations are defined upon first appearance and have added a definition for RMSE (L249, revised MS)**

L219: How did you end up with 96%?

**This was determined through experimentation. We have clarified this in the text:**

**"We impose an experimentally determined heuristic constraint that the alignment must result in more than 96% of the area of the child being within its intersection with the parent" (L244-245, revised MS)**

Evaluation:

L272: Source of traccuracy?

**We have added a link to the traccuracy source code:**

**We used the *traccuracy* python package (github.com/live-image-tracking-tools/traccuracy)" (L309, revised MS)**

Results:

L308: Shortly explain the threshold parameters.

L316-317: Shortly explain the parameters and their meaning.

**We have added a paragraph to the methods section summarising these parameters and their meanings which we hope makes this results section clear:**

**"We thus have five tuneable parameters within the generational linking stage. Sigma is the standard deviation of the gaussian used to model spatial source probabilities while probability threshold determines the probability above which the location of a candidate parent is accepted for consideration in generational linking (see Appendix 1). Proportional overlap is the proportion of the child's area that must fall within the parent for the match to be deemed valid, time buffer is the maximum number of timesteps over which matches are considered and sub-section lengths is the number of vertices used to calculate DTW distances." (L287-293, revised MS)**

Discussion:

: What does A0 – A4 mean in terms of numbers? What is the minimum size an iceberg needs to have to still be tracked back reliably?

We have added corresponding areas to the discussion of size classes:

**"Our custom metrics derived to support expected scientific downstream applications (RP and RAP) show that we successfully track the vast majority of large icebergs (classes A3 and A4, >10 km$^2$) such that we can correctly identify their source. For smaller icebergs (A0-A2), that ability declines, although for A1 (0.1-1 km$^2$) and A2 (1-10 km$^2$) sizes, moderate performance is still achieved" (L415-418, revised MS)**

L374-379: The relation to calving behaviour may only work for glaciers calving large icebergs. For many glaciers in Greenland calving small icebergs at a high temporal frequency the algorithm might struggle.

**This is true, although for smaller icebergs from high frequency calving events their persistence across large distances once outside the fjord is likely to be limited compared to the large tabular icebergs, so the glacier's distal effects on freshwater inputs are likely to be less dependent on iceberg dynamics. We have added a caveat to the text:**

**"We can relate, on average, over 90% of the area of icebergs back to their source when tracklets end. This implies that we are capturing the spatial distribution of most of the ice volume following large calving events (likely greater than the RAP value due to the 3-dimensional geometry of icebergs (Sulak et al., 2017)) and are able to attribute it to particular ice shelves or glaciers in situations where they calve large icebergs." (L427-430, revised MS)**

L396-398: Great idea, but this would probably only work for floating tongues with large icebergs?

**This may well be the case for the current spatially coarse and temporally sparse source of data (CI2D3). For smaller, more frequent calving behaviours a higher frequency and higher resolution initial data source would be required, allowing the changes in calving front and resulting small icebergs to be properly resolved. If this were available, for example from terrestrial or ship-borne radar observations in a specific fjord, then the tracking method should remain applicable with some tuning.**

Discussion & Conclusion:

You tested the approach only for the area around Petermann Glacier and although I think the method is transferable for other regions, you haven't tested it. Also, many glaciers in Greenland do not calve large tabular icebergs and the method would first have to be tested for such areas to see if the approach also works for this kind of icebergs. I agree that this is out of the scope for this manuscript but it should be clearly stated.

**We have added an acknowledgement of this fact towards the end of the discussion along with a recently published dataset that may be useful for wider evaluation.**

**"We have evaluated our approach for the CI2D3 dataset but further work is required to evaluate its generalisability to data from other sources and regions, including other areas of Greenland with differing calving regimes and for Antarctic icebergs (Guan et al., 2025)." (L501-503, revised MS)**

Figure 1:

Would it be possible to use different colours for the different years? Then it would be easier to identify how many observations per year exist. I am not sure I understand (c), is it all the same iceberg?

We like the idea of showing time the distribution of observations. We tried to update the figure with different colours according to calving year but, because of the scale of the map relative to the size of the icebergs, and the superposition of later icebergs over earlier ones, this visualisation skews the apparent distribution of observations towards later years and is therefore potentially misleading. We have therefore chosen to leave the iceberg outlines black to avoid confusion. Regarding panel C, this shows some of the fragmentations undergone by an iceberg (initial identity 1167), following the branches containing the largest fragment. We believe that this should be clear in the context of the manuiscript as a whole, but have added to the caption:

"(c) Schematic of partial lineage tree representing the fragmentation of an iceberg (ID 1167) within the CI2D3 Database, following the branches containing the largest fragment at each division.Colours of branches correspond to the iceberg outlines on the right, numbers denote iceberg ID." (caption, Figure 1)

Figure4:

This figure is hard to understand. Maybe it would help to plot the tracks on a map and use colors to represent time?

We have updated figure 4 with 2-D maps, along with its caption and description in the text. We have used colours to denote whether arcs and generational linkages are TP, FP or FN, so were unable also to use colours to signify time. We hope the revised figure and description provides more insight and clarity.

"Figure 1 shows examples of lineages reconstructed using our method are shown for two timepoints (t=341 in main panel and t=370 in inset) to illustrate correct tracks and various possible failure modes. Five points of interest (A-E) are marked. A shows a fragmentation event that is identified by white circles on both panels to aid orientation. This event produced two FP generational linkages that also imply two FN arcs. B marks a single FN arc in an otherwise long and correct track for a small iceberg. C marks the fragmentation shown in more detail in Error! Reference source not found.c(iii) where three children are correctly matched and one missed. D marks a correctly tracked fragmentation into two children and E shows successive failures (both FP and FN) in the track of a very small iceberg" (L360-366, revised MS)

"Figure 2: Example tracks reconstructed from the CI2D3 dataset prior to two timepoints (t=341 in main panel, t=370 for inset).  True positive (TP), false positive (FP) and false negative (FN) arcs and generational links are shown. Point A (white circle) denotes a fragmentation resulting in one TP generational link and two FP generational links. This event is identified in both pan els for orientation purposes. B denotes a FN link in an otherwise long correct track, C corresponds to the fragmentation shown in Error! Reference source not found.c(iii), D marks a correctly tracked fragmentation, E denotes a very small, poorly tracked iceberg with both FP an FN arcs Greenland elevation data from GIMP-DEM 90 (Howat et al., 2014)." (caption, Figure 4)

Once again, we are grateful to the reviewer for their insight and assistance in improving our manuscript. We hope that we have fully addressed all their questions.